# Semi-Supervised Hypothesis Testing by Betting on Predictions

**Yaniv Tenzer** [* 1]  **Elad Tolochinsky** [* 1]  **Yaniv Romano** [1 2]

## Abstract

We introduce a testing-by-betting framework that leverages predictions on unlabeled data to enhance the power of sequential hypothesis testing. Given limited samples from the joint distribution of $(X, Y)$, and additional unlabeled samples from the marginal of $X$, we ask how unlabeled data can be used to hypothesize about the distribution of $Y$, and the conditional distribution of $Y \mid X$. We introduce an e-statistic and use it to construct a sequential test. Under standard distributional assumptions—label shift or concept shift—we establish that the test is anytime valid. Furthermore, we show that for binary data, the e-statistic has non-trivial power. Crucially, our approach retains these properties even when the underlying predictions are inaccurate. Through simulations and applications to large language models evaluation, we demonstrate power gains over baseline approaches, including prediction-powered inference. These gains persist even with relatively limited unlabeled data and when predictions have low accuracy due to weak correlation between $X$ and $Y$.

## 1. Introduction

Hypothesis testing for an unknown parameter is a foundational problem in statistics, with applications spanning scientific research and modern machine learning. As a concrete example, consider evaluating a newly trained large language model (LLM): one may ask whether the candidate model achieves higher accuracy than an existing baseline. Let $Y$ be a random variable whose distribution is parameterized by an unknown parameter $\theta_Y$; in this example, $\theta_Y$ corresponds to the candidate model's accuracy. Let $\theta_Y^{\text{null}}$

*Equal contribution [1] Department of Computer Science, Technion – Israel Institute of Technology [2] Department of Electrical and Computer Engineering, Technion – Israel Institute of Technology. Correspondence to: Yaniv Tenzer <yanivt@technion.ac.il>, Elad Tolochinsky <elad.t@cs.technion.ac.il>.

*Proceedings of the $43^{rd}$ International Conference on Machine Learning*, Seoul, South Korea. PMLR 306, 2026. Copyright 2026 by the author(s).

denote a reference null value, e.g., the baseline model's accuracy. We thus consider the one-sided hypothesis test

$$(\mathcal{P}_1) \quad H_0 : \theta_Y = \theta_Y^{\text{null}} \quad \text{versus} \quad H_1 : \theta_Y > \theta_Y^{\text{null}} .$$

A closely related problem is testing whether a *conditional* distribution deviates from a specified reference null. For example, imagine a pharmaceutical company that is interested in testing whether the efficacy of a new treatment $Y$ improves for at least one subgroup of $X$ (e.g., gender), relative to a standard reference treatment. Suppose the conditional distribution $Y \mid X = x$ is parameterized by $\theta_{Y \mid X=x}$; in this example, $\theta_{Y \mid X=x}$ captures the treatment's efficacy within subgroup $X = x$. With this in place, we consider the following hypothesis test:

$$(\mathcal{P}_2) \quad H_0 : \theta_{Y \mid X} = \theta_{Y \mid X}^{\text{null}} \text{ for all } X = x$$
$$H_1 : \theta_{Y \mid X} > \theta_{Y \mid X}^{\text{null}} \text{ for at least one } X = x.$$

We refer to $H_0$ and $H_1$ as the null and alternative hypothesis, respectively, and denote by $H_0(\mathcal{P}_1), H_0(\mathcal{P}_2)$ the null hypothesis of $\mathcal{P}_1, \mathcal{P}_2$, respectively. Similarly, $H_1(\mathcal{P}_1), H_1(\mathcal{P}_2)$ denote the alternative hypothesis of $\mathcal{P}_1, \mathcal{P}_2$, respectively.

Common testing procedures aim at testing the null hypothesis by collecting data and quantifying the probability of the observed data under the null. Conceptually, the quantified probability serves as evidence against the null. Two desired properties of any statistical test are *validity* and *power*. Formally, we say that a test is valid at level $\alpha \in (0, 1)$ if the test erroneously rejects the null hypothesis, when it is actually true, with probability at most $\alpha$. This kind of erroneous rejection is known as type-I error. As mentioned, the second desired property is *power*. Formally, we say that a test has power $\beta \in (0, 1)$ if it rejects the null with probability $\beta$, when the alternative is true. A core challenge in statistical inference is to design tests that are both valid and powerful.

Traditional hypothesis testing procedures require the sample size to be specified in advance, before any data are collected. Consequently, the sample size cannot be adjusted as data are collected and evidence against the null is accumulated. Unfortunately, this can lead to inefficient data collection, with many samples being unnecessarily allocated to easy testing problems, or too few samples being allocated to

more challenging settings, causing the test to terminate before obtaining sufficient evidence against the null. In contrast, sequential testing procedures can allow the null hypothesis to be tested dynamically at each stage of data collection, enabling the test to stop as soon as sufficient evidence has accumulated. We refer to such tests as ones that attain *anytime valid* type-I error control. In this context, a powerful test is a test that under the alternative quickly rejects the null.

Most recent online testing approaches are based on the general testing by betting framework, introduced in (Shafer, 2021). Under this paradigm, a fictitious bettor sequentially bets against the null, with the bettor's wealth serving as a running measure of statistical evidence. At each step, the wealth increases or decreases according to a payoff function evaluated on fresh samples. To obtain a valid test, the payoff function must be chosen so that the resulting betting game is fair: under the null, the expected wealth of the bettor should not exceed one. When the alternative is true, a well-designed betting strategy can yield rapid growth of wealth, thereby providing high testing power.

Unlike the classic sequential settings, whereby at each step we observe only paired $(X, Y)$ samples, here we consider the setting in which at each step $t$, we observe a batch of $n$ i.i.d labeled samples $D_t = \{(X_i, Y_i)\}_{i=1}^n$, and additional $N$ i.i.d unlabeled $X$-samples $\tilde{\underline{X}}_t = \{\tilde{X}_i\}_{i=1}^N$. Conceptually, we think of the acquisition of $X$-samples as cheaper than the acquisition of $Y$-samples. We emphasize that this setting matches many modern applications. For example, consider again the task of LLM evaluation. As obtaining human annotations, $Y$, can be expensive, we typically resort to LLM annotations, $X$, which are commonly cheaper to obtain. In this setup, $Y$ samples are typically scarce, while $X$ samples are abundant. A crucial question is therefore: *how can the extra unlabeled $X$-samples be used to enhance power, while still ensuring type-I error control?*

**Preview of Our Method and Main Contributions**

Our first key contribution—presented in section 3—is a rigorous construction that leverages unlabeled data to enhance test power by *betting on predictions*. At each step of the test, we fit a predictive model using fresh labeled data $D_t$ and apply it to assign predictions to unlabeled samples $\tilde{\underline{X}}_t$. We then show how to construct two valid sequential tests—for $\mathcal{P}_1$ and $\mathcal{P}_2$—that utilize these predictions.

Importantly, a priori, any sequential test that uses the observed data $D_t, \tilde{\underline{X}}_t$, may only be used to infer about the joint distribution of $(X, Y)$. However, we wish to infer either about $Y$ or about $Y \mid X$. Therefore, a fundamental question is: when can one draw conclusions about $Y$ and $Y|X$ using $D$ and $\tilde{\underline{X}}$ samples? We formalize identifiability conditions that facilitate such inference, and show that under

these conditions, the resulting tests attain anytime type-I error control in finite samples, regardless of the accuracy of the predictive model.

Our second key contribution—presented in section 4—is a power analysis of our test statistic. Specifically, we focus on the contingency tables setup, i.e., $X$ and $Y$ are binary variables. We note that this is a fundamental structure in statistical analysis that is the subject of ongoing research (Turner & Grünwald, 2023) and is prevalent in many popular real-life scenarios such as clinical trials (Polack et al., 2020), A/B testing (Kohavi et al., 2022), and bias analysis (Obermeyer et al., 2019). Our analysis reveals that even inaccurate models suffice to yield non-trivial power test statistics.

In simulations and real-world experiments, we demonstrate that our approach shows meaningful improvement in power even when $X$ and $Y$ have low correlation, and the number of unlabeled samples, $N$, is small. Experiments also demonstrate that our method can enhance power even when applied alongside strong baselines that utilize unlabeled data. Code is available at `https://github.com/elad-tolo/betting_on_predictions`

## 2. Background and related work

### 2.1. Testing by betting: e-value and e-process

An e-statistic, $e$, is any non-negative function of the data $D$, such that $\mathbb{E}_{H_0}[e(D)] \leq 1$. A realization of an e-statistic will be referred to as an e-value. By Markov's inequality, $P_{H_0}(e(D) > 1/\alpha) \leq \alpha$. Therefore, the type-I error of the test $\mathbb{1}\{e(D) \geq 1/\alpha\}$ is bounded by $\alpha$, and large e-values constitute evidence against the null hypothesis. An e-statistic, $e$, is said to have non-trivial power against an alternative $H_1$ if $\mathbb{E}_{H_1}[e(D)] > 1$.

In a sequential setting, we observe a stream of data $D_1, D_2, \ldots, D_t$ for $t \in \mathbb{N}$. We define $D_{-t} = \bigcup_{i=1}^{t-1} D_i$ and let $\mathcal{F} = (\mathcal{F}_t)_{t \in \mathbb{N}}$ be the natural filtration of the data. Conditional e-statistics are random variables $e(D_t, D_{-t})$, which satisfies $\mathbb{E}_{H_0}[e(D_t, D_{-t}) \mid \mathcal{F}_{t-1}] \leq 1$, for all $t \in \mathbb{N}$, where for $t = 1$, the expectation is supposed to be read unconditionally. Intuitively, the conditional e-statistic at time $t$ measures the evidence in round $t$ against $H_0$ given past data. The cumulative product of these conditional e-statistics $S_t = \prod_{i=1}^t e(D_i, D_{-i})$ is an *e-process* and measures the total accumulated evidence against the null hypothesis. Indeed, by Ville's inequality, for any $\alpha > 0$, $P_{H_0}(\exists t: S_t > 1/\alpha) \leq \alpha$ (Ville, 1939). A sequential test can thus be defined by monitoring $S_t$ and rejecting the null if it exceeds $1/\alpha$. For a complete introduction to statistical testing using e-values, see (Ramdas & Wang, 2025).

## 2.2. Related work

Our method is inspired by the popular construction of soft-rank e-value (Ramdas & Wang, 2025) and by the recent work of (Grünwald et al., 2024) that utilizes a similar construction to test conditional independence. Similar to these works, we sample additional data from the null to construct our e-statistic. However, unlike these works, our construction involves predictions derived from unlabeled data.

Prediction-powered inference (PPI) is a recent work that tackles the challenge of using unlabeled data for inference (Angelopoulos et al., 2023a;b). This semi-supervised framework constructs unbiased estimates and valid confidence intervals for statistical parameters of the distribution of $Y$, such as their mean or quantiles, using predictions assigned to the unlabeled $\tilde{X}$-s. Several recent works used this framework, including applications for risk-control (Einbinder et al., 2025) and hypothesis testing (Csillag et al., 2025; Kilian et al., 2026). The advantage of PPI is mainly evident when $N$ is large, and the predictive model is fairly accurate. In such cases, the variance of the estimate is reduced compared to an estimate that only uses the labeled data, which results in a more powerful test. In contrast, our approach can gain power even when $N$ is small, and the predictive model is inaccurate. Another key difference is that in PPI, bets are placed on labeled data, while in our approach, bets are placed directly on predictions. We refer the reader to Appendix E where we formally define a sequential test based on PPI and discuss these differences in detail.

## 3. Our Method: Betting on Predictions

The following notations will be useful in the sequel. We denote the joint distribution of $X, Y$ under the null with $P_{XY}^{\text{null}}$, and the marginal distribution of $X$ by $P_X^{\text{null}}$. We let $P^{\text{null}}(D, \tilde{X})$ be the distribution of the observed data under the null. As a first step towards a sequential test, we consider the offline case where we are given a single dataset $D$ of labeled data and a single unlabeled dataset $\tilde{X}$.

### 3.1. Batch Mode: Imputed E-Statistic

Let $\mathcal{A}$ be a learning algorithm. We run $\mathcal{A}$ to fit a predictive model on the labeled data $D$, and use the resulting model to predict to the unlabeled data points. We denote by $\mathcal{A}[D, \tilde{X}] = (\mathcal{A}[D, \tilde{X}_1], \ldots, \mathcal{A}[D, \tilde{X}_N])$ the vector of predictions. Importantly, the same model is used to predict the labels of all unlabeled data points.

Next, we introduce a novel e-statistic construction that gains evidence against the null from predicted labels:

$$e[D, \tilde{X}] = \frac{K(\mathcal{A}[D, \tilde{X}])}{\mathbb{E}_{P^{\text{null}}(D, \tilde{X})}[\mathcal{A}[D, \tilde{X}])]}, \qquad (1)$$

where $K : \mathbb{R}^N \to \mathbb{R}^+$ is a non-negative function. Concretely, for a binary $\tilde{Y}$, in our experiments we use $K(\tilde{y}) = \sum_{i=1}^N \exp(\gamma \tilde{y}_i)$ for some hyper-parameter $\gamma \in \mathbb{R}^+$.

Unfortunately, calculating (1) can be infeasible due to the expectation in the denominator. Luckily, we can bypass this by sampling $M$ independent data sets from the null, $(D^1, \tilde{X}^1), \ldots, (D^M, \tilde{X}^M)$. Denote $(D^0, \tilde{X}^0) = (D, \tilde{X})$ and $\tilde{Y}^i = \mathcal{A}[D^i, \tilde{X}^i]$. We define the finite-sample imputed e-statistic as follows:

$$\breve{e}[D, \tilde{X}] = \frac{(M+1)K(\tilde{Y}^0)}{K(\tilde{Y}_0) + \sum_{i=1}^M K(\tilde{Y}^i)}. \qquad (2)$$

Two notes are in place: firstly, $\breve{e}[D, \tilde{X}]$ follows the structure of a soft-rank e-value (Ramdas & Wang, 2025). Indeed, if $K(\cdot)$ also preserves some notion of order, then $\breve{e}[D, \tilde{X}]$ is a soft-rank e-value. We discuss this in detail in Section 4. Secondly, since $(D^1, \tilde{X}^1), \ldots, (D^M, \tilde{X}^M)$ are sampled i.i.d from the null, under the null hypothesis, they are exchangeable with the observed data $(D^0, \tilde{X}^0)$. Therefore, $\breve{e}[D, \tilde{X}]$ is designed to test exchangeability of $(D^0, \tilde{X}^0)$ with $(D^1, \tilde{X}^1), \ldots, (D^M, \tilde{X}^M)$. However, our goal is to infer about $Y$ ($\mathcal{P}_1$) or about $Y \mid X$ ($\mathcal{P}_2$). Thus, we must derive conditions under which testing exchangeability is equivalent to testing $\mathcal{P}_1$ or $\mathcal{P}_2$. We next address this problem.

### 3.2. Identifiability Conditions and Validity

Starting with $\mathcal{P}_1$, we aim to identify conditions under which rejecting the equality in distribution hypothesis, $(D^0, \tilde{X}^0) \overset{d}{=} (D^i, \tilde{X}^i)$, is equivalent to rejecting $H_0(\mathcal{P}_1)$. Observe that:

$$P^{\text{null}}(D, \tilde{X}) = \prod_{i=1}^n P_{XY}^{\text{null}}(x_i, y_i) \prod_{i=1}^N P_X^{\text{null}}(\tilde{x}_i).$$

We decompose $P_{XY}$ as follows: $P_{XY}^{\text{null}}(x_i, y_i) = P_Y^{\text{null}}(y_i) P_{X|Y}^{\text{null}}(x_i|y_i)$. When $P_{X|Y}$ is fixed between the null and observed distributions, the above decomposition implies that a shift in $P^{\text{null}}$ must be attributed to either a shift in $P_Y^{\text{null}}$ or in $P_X^{\text{null}}$. However, if $P_{X|Y}$ is fixed, then a shift in $P_X^{\text{null}}$ must also indicate a shift in $P_Y^{\text{null}}$. Indeed, $P_X(x) = \int P_{X|Y}(x \mid Y = y) P_Y(y) dy$. Thus, if $P_Y$ is not shifted, and $P_{X|Y}$ is fixed, then so is $P_X$, which is a contradiction. We conclude that if $P_{X|Y}$ is fixed, then any observed deviation from $P^{\text{null}}$ must indicate a shift in $P_Y$ and we can use the imputed e-statistics to test $\mathcal{P}_1$. The assumption of a fixed $P_{X|Y}$ is known as a *label shift* setting.

Moving to $\mathcal{P}_2$, our goal is to infer about $P_{Y|X}$. We now decompose $P_{XY}$ as follows: $P_{XY}^{\text{null}}(x_i, y_i) = P_X^{\text{null}}(x_i) P_{Y|X}^{\text{null}}(y_i|x_i)$. When $P_X$ is fixed, then any observed deviation from $P^{\text{null}}$ trivially indicates a shift in $P_{Y|X}$

and thus we can use the imputed e-statistics to test $\mathcal{P}_2$. We note that the assumption of a fixed $P_X$ is known as a *concept shift* setting.

Next, equipped with the above identifiability conditions, we establish the validity of $e[D, \tilde{X}]$ and $\breve{e}[D, \tilde{X}]$. The validity of $e[D, \tilde{X}]$ is immediate; indeed, under label shift setting (concept shift), if $H_0(\mathcal{P}_1)$ $(H_0(\mathcal{P}_2))$ holds, then $(D, \tilde{X}) \sim P^{\text{null}}(D, \tilde{X})$. Thus, $\mathbb{E}_{H_0(\mathcal{P}_1)}[e[D, \tilde{X}]] = 1$ and $\mathbb{E}_{H_0(\mathcal{P}_2)}[e[D, \tilde{X}]] = 1$. Therefore it remains to establish the validity of $\breve{e}[D, \tilde{X}]$:

**Proposition 3.1.** *(Informal)* $\breve{e}[D, \tilde{X}]$ *is a valid e-statistic with respect to both $\mathcal{P}_1$ and $\mathcal{P}_2$.*

A formal statement and proof can be found in Appendix B.1. Importantly, Proposition 3.1 holds regardless of the learning algorithm $\mathcal{A}$, the accuracy of the resulting model, the number of labeled samples, $n$, or unlabeled samples $N$. We next use this result to derive a robust sequential test.

### 3.3. Sequential E-process

We begin by updating our previous notations to match the sequential nature of our setting. At step $t$, we observe a fresh batch of labeled and unlabeled data $(D_t, \tilde{X}_t)$. Let $(D_t^1, \tilde{X}_t^1), \ldots, (D_t^M, \tilde{X}_t^M)$ be $M$ independent data sets drawn from the null at step $t$, and we denote $(D_t^0, \tilde{X}_t^0) = (D_t, \tilde{X}_t)$. Aligned with previous notations, let $\tilde{Y}_t^i = \mathcal{A}[D_t^i, \tilde{X}_t^i]$ be the predicted labels of $\tilde{X}_t^i$ made by a model that is fitted on $D_t^i$.

We define the *conditional* finite-sample e-statistic, $\breve{e}[D_t, \tilde{X}_t]$, similarly to $\breve{e}[D, \tilde{X}]$ (2), however, now it depends on the historical data through $K(\cdot)$. To emphasize this, we index $K(\cdot)$ by $t$: $K_t(\cdot)$. Importantly, at time $t$, the choice of $K_t(\cdot)$ is based on the historical data, $D_{-t}$, merely. For example, $K_t(\cdot)$ might be parametrized by some parameter $\gamma_t$ that is tuned in an online manner. It is easy to show that under the same conditions as in Proposition 3.1, $\breve{e}[D_t, \tilde{X}_t]$ is a valid conditional e-statistic.

**Proposition 3.2.** *(Informal)* $\breve{e}[D_t, \tilde{X}_t]$ *is a valid conditional e-statistic with respect to both $\mathcal{P}_1$ and $\mathcal{P}_2$.*

A formal statement along with the proof is in Appendix B.2.

Next, let $S_t = \prod_t \breve{e}[D_t, \tilde{X}_t]$. Assuming a label shift setting (concept shift), under $H_0(\mathcal{P}_1)(H_0(\mathcal{P}_2))$, $S_t$ is an e-process. Thus, under both settings, by Ville's inequality, we get type-I error control:

**Corollary 3.3.** *For any $\alpha > 0$, $P_{H_0(\mathcal{P}_i)}(\exists t : S_t > 1/\alpha) \leq \alpha$, where $i \in \{1, 2\}$.*

Crucially, as before, Proposition 3.2 and Corollary 3.3 hold regardless of the choice of $\mathcal{A}$, $N$, $n$, and the accuracy of predictions.

Finally, as we show in Appendix C and Appendix D, when $X$ and $Y$ are binary variables, $\mathcal{P}_1$ and $\mathcal{P}_2$ can be extended to composite null testing, while the test remains valid.

### 3.4. Robustness

In practice, $P^{\text{null}}(D, \tilde{X})$ is often unknown, and thus samples are drawn from some approximated distribution $\hat{P}_t^{\text{null}}(D, \tilde{X})$. Note that the practitioner is allowed to update her approximation of $P^{\text{null}}(D, \tilde{X})$ at each step (for example, if given access to more recent data, etc.). Let $\hat{P}_t^{\text{null}}(D, \tilde{X})$ be the approximated distribution used at the $t$-th step, and let the approximated e-statistic at time $t$ be:

$$\hat{e}[D_t, \tilde{X}_t] = \frac{K(\tilde{Y}_t)}{\mathbb{E}_{\hat{P}_t^{\text{null}}}[K(\tilde{Y}_t)]}, \tag{3}$$

Let $(D_1, \tilde{X}_1) \stackrel{d}{=} \ldots \stackrel{d}{=} (D_t, \tilde{X}_t)$ be independent identically distributed copies of $(D, \tilde{X})$. We let $P_{\otimes_t}^{\text{null}}$ be their joint distribution under the null: $P_{\otimes_t}^{\text{null}}((D_1, \tilde{X}_1), \ldots, (D_t, \tilde{X}_t)) = \prod_{i=1}^t P_i^{\text{null}}(D_i, \tilde{X}_i)$ and define $\hat{P}_{\otimes_t}^{\text{null}}$ analogously. The following result states that at each time step $t$, using the approximated e-statistic inflates the type-I error by the total-variation distance between $P_{\otimes_t}^{\text{null}}$ and $\hat{P}_{\otimes_t}^{\text{null}}$. The proof can be found in Appendix B.2.

**Theorem 3.4.** *Let $K(\cdot)$ be a strictly positive score function. Then, for any $t \in \mathbb{N}$:*

$$P\left(\exists t' \leq t : \prod_{i=1}^{t'} \hat{e}[D_i, \tilde{X}_i] \geq \frac{1}{\alpha}\right) \leq \alpha + d_{TV}(P_{\otimes_t}^{\text{null}}, \hat{P}_{\otimes_t}^{\text{null}}).$$

The above theorem is non-trivial when the excess rejection rate, $d_{TV}(P_{\otimes_t}^{\text{null}}, \hat{P}_{\otimes_t}^{\text{null}})$, is sufficiently small. Deriving a general upper bound for the excess rejection rate is an open question. Nevertheless, in this work we focus on binary variables $X, Y$ and it can be shown that if we have access to sufficient number of samples from the null, $d_{TV}(P_{\otimes_t}^{\text{null}}, \hat{P}_{\otimes_t}^{\text{null}})$ can be made small with high probability. We discuss this further in Appendix B.3. Note that a qualitatively similar worst-case bound on rejection rate appears in (Grünwald et al., 2024), specifically in the context of conditional independence testing using e-processes.

Until now, we have focused on the validity of the imputed e-statistic as well as on the test's validity and its robustness. Note that we were able to obtain these results with minimal assumptions about the score function $K(\cdot)$. Concretely, our only assumption is that $K(\cdot)$ is a positive function. In addition, we did not commit to any specific learning algorithm $\mathcal{A}$. Next, we focus on the power of the imputed e-statistic, and explore conditions under which it has non-trivial power against the alternative.

# 4. Power Analysis in the Binary Case

We consider the case where $X$ and $Y$ are Bernoulli variables and show that under both label and concept shift, $e[D, \tilde{X}]$ (1), $\breve{e}[D, \tilde{X}]$ and the conditional e-statistic $\breve{e}[D_t, \underline{\tilde{X}}_t]$ (2) are powered against the alternative. For concreteness, as a score function we take $K(\underline{y}) = \sum_{i=1}^{N} \psi(\tilde{y}_i)$, where $\psi(\cdot)$ is a positive increasing function (e.g. $\psi(y_i) = \exp(\gamma \tilde{y}_i)$). Note that under this choice of $K(\cdot)$, $\breve{e}[D, \tilde{X}]$ is now a soft-rank e-statistic, as $K$ is monotone increasing and thus order preserving (Ramdas & Wang, 2025); see Appendix A.2.

## 4.1. Non-Trivial Power under Label-Shift

We focus on $\mathcal{P}_1$, which, as discussed, under the label-shift regime can be tested with our imputed e-statistic. Recall that in this setting, $P_{X|Y}$ is fixed. We next show that with a particular choice of the learning algorithm $\mathcal{A}$, the imputed e-statistic has non-trivial power. Concretely, for $\tau \in (0, 1)$, we consider the threshold classifier:

$$\mathcal{A}_\tau[D, \tilde{x}] = \mathbb{1}\{\hat{P}(Y = 1 \mid \tilde{x}) > \tau\}, \qquad (4)$$

where

$$\hat{P}(Y = 1 \mid \tilde{x}) = \frac{\hat{P}_D(Y = 1)\hat{P}(\tilde{x} \mid Y = 1)}{\sum\limits_{y \in \{0,1\}} \hat{P}_D(Y = y)\hat{P}(\tilde{x} \mid Y = y)}. \quad (5)$$

In (5) above $\hat{P}_D(Y = 1)$ is the empirical mean of $Y$ in $D$, and $\hat{P}(X|Y = y)$ are the empirical estimates of $P(X \mid Y = y)$ calculated using the additional null data. Note that since $P_{X|Y}$ is fixed, by Slutsky's (Slutsky, 1925), (5) yields a consistent estimator of $P(Y = 1|\tilde{x})$ under the alternative.

Recall that an e-statistic $e(D)$ has non-trivial power against alternative distribution $Q$ if $\mathbb{E}_Q[e(D)] > 1$. With this in place we establish the following result:

**Theorem 4.1.** *(Informal) Take $\mathcal{A}_\tau[D, \tilde{x}]$ as in (4). Then, under the label-shift regime, $e[D, \tilde{X}], \breve{e}[D, \underline{\tilde{X}}]$, and $\breve{e}[D_t, \underline{\tilde{X}}_t]$ all have non trivial power against $H_1(\mathcal{P}_1)$.*

A formal statement and the proof can be found in Appendix B.4.

## 4.2. Power of Imputed E-Statistics under Concept Shift

We next focus on $\mathcal{P}_2$, which, as was shown, under the concept shift regime, can be tested with our imputed e-statistic. Recall that in this setting, $P_X$ is fixed. Here $\mathcal{A}$ classifies a new observation $\tilde{x}$ by drawing a sample from the empirical posterior probability:

$$\mathcal{A}[D, \tilde{x}] \sim Ber(\hat{P}(Y = 1 \mid \tilde{x})), \qquad (6)$$

where the posterior probabilities $P(Y = 1 \mid \tilde{x})$ are estimated from $D$. We refer to this classifier as the *Bayes'*

classifier. Note that with the specification of $K$ as above, establishing non-trivial power result requires further assumption about the alternative distributions. Indeed, as $K(\cdot)$ is monotone increasing, we expect the e-statistic to have power against right-shifted alternatives, i.e., alternatives for which $\theta_{Y|X} \geq \theta_{Y|X}^{\text{null}}$, for all $x$. Thus, we consider the following subset of $H_1(\mathcal{P}_2)$:

$$\theta_{Y|X} > \theta_{Y|X}^{\text{null}} \text{ for at least one } X = x, \text{ and}$$

$$\theta_{Y|X} = \theta_{Y|X}^{\text{null}} \text{ for every other } x.$$

We denote this subset with $H_1(\mathcal{P}_2^+)$. With this in place, we establish the following result:

**Theorem 4.2.** *(Informal) Take $\mathcal{A}[D, \tilde{x}]$ as above. Then, under the concept shift regime, $e[D, \tilde{X}], \breve{e}[D, \underline{\tilde{X}}]$, and $\breve{e}[D_t, \underline{\tilde{X}}_t]$ all have non trivial power against $H_1(\mathcal{P}_2^+)$.*

The formal statement and the proof can be found in Appendix B.5. Finally, note that designing powerful tests with respect to a given alternative requires an adequate choice of $K(\cdot)$. Indeed, we empirically demonstrate this in Appendix F.4. There, we show how an inappropriate choice of $K$ might lead to poor results.

# 5. Simulations

Focusing on the binary setting where $X$ and $Y$ are Bernoulli variables, we empirically evaluate the benefit of our imputed e-process compared to popular baselines. We consider different regimes, in which we vary the number of unlabeled points $N$ and levels of correlation between $X$ and $Y$.

## 5.1. Baselines

**Likelihood Ratio Test**  We compare our method to the LR e-process (Wasserman et al., 2020) . We denote by $e_{\text{LR}}^Y, e_{\text{LR}}^X,$ and $e_{\text{LR}}^{Y|X}$ the sequential likelihood ratio e-statistics with respect to the marginal distribution of $Y$, $X$, and the conditional distribution $Y|X$, respectively. Under the label-shift setting, $P_X$ and $P_Y$ may shift; thus, we instantiate both $e_{\text{LR}}^Y$ and $e_{\text{LR}}^X$. For concept shift, $P_X$ is fixed, rendering $e_{\text{LR}}^X$ irrelevant. However, the $H_1(\mathcal{P}_2^+)$ implies that there exists at least one $x$ such that $\theta_{Y|X=x} > \theta_{Y|X=x}^{\text{null}}$, while for other values $\theta_{Y|X=x} = \theta_{Y|X=x}^{\text{null}}$. Consequently, $\theta_Y > \theta_Y^{\text{null}}$, and we therefore consider both $e_{\text{LR}}^Y$ and $e_{\text{LR}}^{Y|X}$.

**PPI**  To compare our method against a test that leverages both labeled and unlabeled data, we employ PPI to design a sequential test for the marginal of $Y$, we denote this e-process $e_{\text{PPI}}$. Importantly, when instantiating the PPI test, we optimize the test's hyper-parameters to guarantee exponential growth of the e-process. See Appendix E for a complete definition of the test along with proof of validity and additional direct comparisons to our method.

## 5.2. Unified e-process

Our goal is not to replace existing methods, but to leverage additional unlabeled data to strengthen them. Furthermore, given that different methods offer different benefits, depending on the setting, and that their "ranking" in terms of power can change between steps, it is generally impossible to know a priori which method will be most powerful. Therefore, we propose a combined process designed to maximize the growth-rate of the resulting e-process.

Formally, let $e_{1,t}, \ldots, e_{k,t}$ be sequential e-variables, it is easy to verify that any convex combination of those e-variables is a sequential e-variable. We define the combined e-variable:

$$\text{conv}(e_{1,t}, \ldots, e_{k,t}, \underline{a}_t) := \sum_{i=1}^{k} a_{i,t} e_{i,t}, \qquad (7)$$

Thus, at time $t$, let $\underline{a}_t \in \mathbb{R}^k$ such that $\sum_{i=1}^{k} a_{t,i} = 1$ and $a_{t,i} \geq 0$ for $i \in \{1, \ldots, k\}$. Note that $\text{conv}(e_{1,t}, \ldots, e_{k,t}, \underline{a}_t)$ is a sequential e-variable as long as $\underline{a}_t$ respects the filtration. Following Waudby-Smith et al. (2025), we optimally set $a_t$ with the Exponentiated Gradient algorithm (Kivinen & Warmuth, 1997) which is a universal portfolio selection algorithm (Helmbold et al., 1998).

## 5.3. Experimental Setup

To instantiate the imputed e-process, we employ the function $K(\tilde{y}) = \sum_{i=1}^{N} \exp(\gamma \tilde{y}_i)$. We note that the parameter $\gamma$ may vary between steps, provided it respects the filtration. Concretely, we set $\gamma$ to optimize the e-power of the imputed e-process using online learning; specifically, we use AdaGrad (Duchi et al., 2011).

To set $M$, the number of exchangeable datasets drawn from the null, we empirically investigate the impact of $M$ on the power of the test. We observed that power generally increases asymptotically with $M$, thus we set $M = 128$ in all simulations and experiments. The full details can be found in Appendix F.3.

In all settings, $X$ and $Y$ are Bernoulli variables. To simulate settings where data scarcity is a concern, and the use of unlabeled data can be attractive, at each step, we draw $n = 15$ labeled samples and $N$ unlabeled samples from a distribution that is relatively close to the null distribution. We simulate each sequential test for 500 steps, and we report the average rejection rate at every step, along with the standard error. We report the average over 500 independent realizations of the data.

## 5.4. Label-Shift

For the null hypothesis, we set $\theta_Y^{\text{null}} = 0.5$ and for the alternative, we use $\theta_Y = 0.52$. We use the threshold classifier (4)

for both the imputed and the PPI e-processes. Under the label shift regime, a shift in $P_X$ can be a powerful indicator of a shift in $P_Y$, and the strength of the signal in $X$ depends on $N$ and on the correlation between $X$ and $Y$. Thus, to study the benefit of our method, we simulate different signal strengths of $X$ by setting $N = 30$ and $N = 135$ and setting the correlations to 0.3 and 0.7.

For each configuration, we plot the power as a function of the step number $t$ of the convex combination of all baselines: $\text{conv}(e_{\text{LR}}^Y, e_{\text{LR}}^X, e_{\text{PPI}})$ and the power of the convex combination of the baselines with our imputed e-process: $\text{conv}(e_{\text{LR}}^Y, e_{\text{LR}}^X, e_{\text{PPI}}, \breve{e}_t)$. Note that any difference in power between the two curves is attributed to our imputed e-process. In addition, to gain intuition about which distribution shift ($P_X$ or $P_Y$) is the most dominant in each setting, we plot both $e_{\text{LR}}^Y$ and $e_{\text{LR}}^X$. Finally, we plot the power curve of the imputed e-process to examine its performance under different settings. In Appendix E.2 we also thoroughly compare to PPI and empirically show that in challenging settings, where the correlation between $X$ and $Y$ is low, our method gains higher power.

Figure 1 presents the results. As can be seen, when the correlation is low (Figures 1a and 1b), our imputed process demonstrates significant power gains. When unlabeled data is scarce (Figure 1a), the convex combination containing our method (blue curve) achieves significantly higher power than the convex combination of the baselines (orange curve). Furthermore, we can observe that for most of the timesteps, the power of our method (red curve) is significantly higher than the power of the most powerful single baseline (green curve). This is due to the scarcity of unlabeled points and low correlation, which render baselines ineffective. When the number of unlabeled points is large (Figure 1b), the gain from our test diminishes, although it remains significant. This is due to the abundance of unlabeled samples, which greatly improves $e_{\text{LR}}^X$.

When the correlation is high, and the number of unlabeled points is low (Figure 1c), we observe that $e_{\text{LR}}^X$ is the most powerful baseline, as expected. It is also evident that after sufficient steps, our process (i) performs on par with $e_{\text{LR}}^X$; and (ii) is capable of utilizing evidence against the null that has not been used by other baselines, thus increasing the power of the combined process. Finally, when both the correlation and the number of unlabeled samples are high (Figure 1d), $e_{\text{LR}}^X$ is the most powerful baseline and its power rapidly grows to one. To conclude, we see that our method is useful even in challenging settings where the correlation is low or $N$ is small.

## 5.5. Concept-Shift

We repeat the same simulation protocol with the following changes: First, under the concept shift regime the

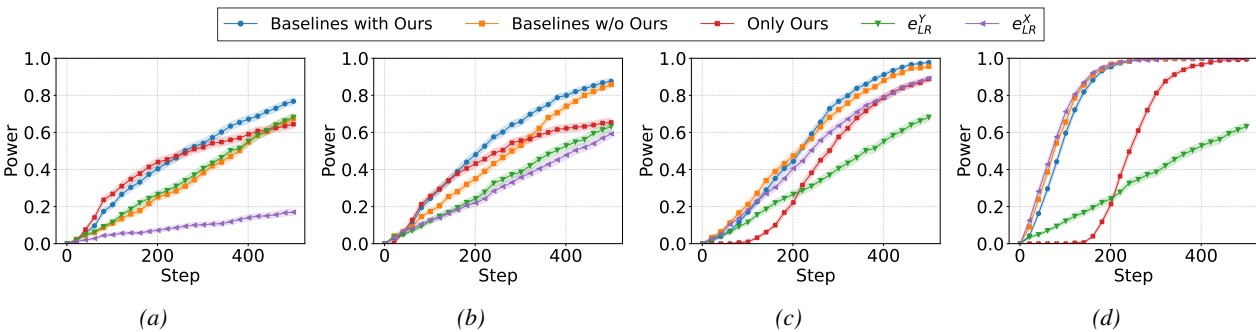

*Figure 1.* Simulation results for the label shift setting. In all settings, the signal strength of the labeled data is fixed and the signal strength of the unlabeled data varies between plots: (a) weak ($N = 30$, low correlation), (b) medium ($N = 135$, low correlation), (c) medium-high ($N = 30$, high correlation), (d) strong ($N = 135$, high correlation).

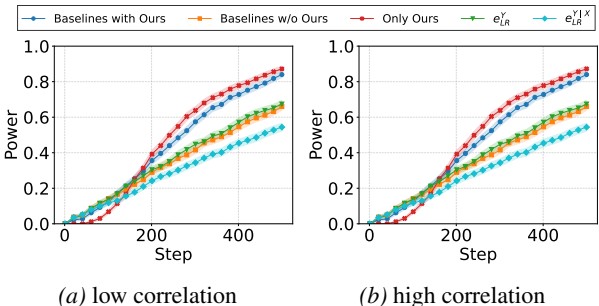

*(a)* low correlation       *(b)* high correlation

*Figure 2.* Simulation results for the concept shift setting.

appropriate baselines are $e_{\text{LR}}^Y$ and $e_{\text{LR}}^{Y|X}$, thus we plot $\text{conv}(e_{\text{LR}}^Y, e_{\text{LR}}^{Y|X}, e_{\text{PPI}})$ and $\text{conv}(e_{\text{LR}}^Y, e_{\text{LR}}^{Y|X}, e_{\text{PPI}}, \breve{e}_t)$. Additionally, since we assume we can freely sample unlabeled data from the null, and in this setting $P_X$ is fixed between the null and the alternative, we only consider large $N$, concretely $N = 135$.

Figure 2 displays the results. Comparing the power of the combined e-process that includes our method (blue curve) with the one that does not include our method (orange curve), we can see that, regardless of the level of correlation, our imputed e-process provides a significant gain in power. In line with the label-shift results, the relative gain is highest when the correlation is low. In both settings, our test (red curve) emerges as the strongest method. This occurs because, under concept shift, our test is the only test that bets on unlabeled data against the conditional null $Y \mid X$.

# 6. Real-World Application: LLM Evaluation

Suppose we have two LLMs: $A$ and $B$, and our goal is to test if model $B$ is more accurate than $A$. We define $Y_A$ and $Y_B$ to be binary random variables indicating a correct answer by the models $A$ and $B$, respectively. For example, for LLM instruction-following tasks, $Y$ can indicate if the LLM's output is helpful; for LLM alignment tasks, $Y$ can indicate if the response is toxic; and for math questions, $Y$ can indicate if the answer is correct. Concretely, we demonstrate our method using math datasets, where $Y$ indicates if the answer to a math question is correct. To implement our method, we assume that we have many samples of $Y_A$, to form the null datasets. This assumption is realistic in cases where model $A$ has been deployed for some time and model $B$ is a new model that we wish to test.

We show how we can use our method with different types of unlabeled data. For example, we can generate synthetic answers using an LLM to compare models $A$ and $B$.

In all experiments, we form a dataset for which the accuracy of models $A$ and $B$ is close. The reason is that any reasonable hypothesis test would quickly reject the null when the performance of the two models greatly differs; arguably, in such settings, existing tests are already powerful, and thus the use of unlabeled data becomes less attractive.

Naturally, in real-world applications, we often have limited data, but our method requires sampling $M$ datasets at each step of the test. To implement the test, we leverage Theorem 3.4 and learn the null distribution from the data. We report the type-I error in section G.5 that validates the robustness of the test.

## 6.1. Data and Models

For the models, we use distillations of DeepSeek-R1 (DeepSeek-AI, 2025) into different families and sizes. Specifically, we use Llama3.1 8B (Grattafiori et al., 2024) as model $A$ and Qwen2.5 7B (Yang et al., 2024) as model $B$. To generate synthetic answers, we use Qwen2.5 14B. For the datasets, we use a union of three popular math datasets: GSM8K (Cobbe et al., 2021), MATH (Hendrycks et al., 2021), and AQUA-RAT (Ling et al., 2017). In Appendix G.1 we provide the details of each dataset.

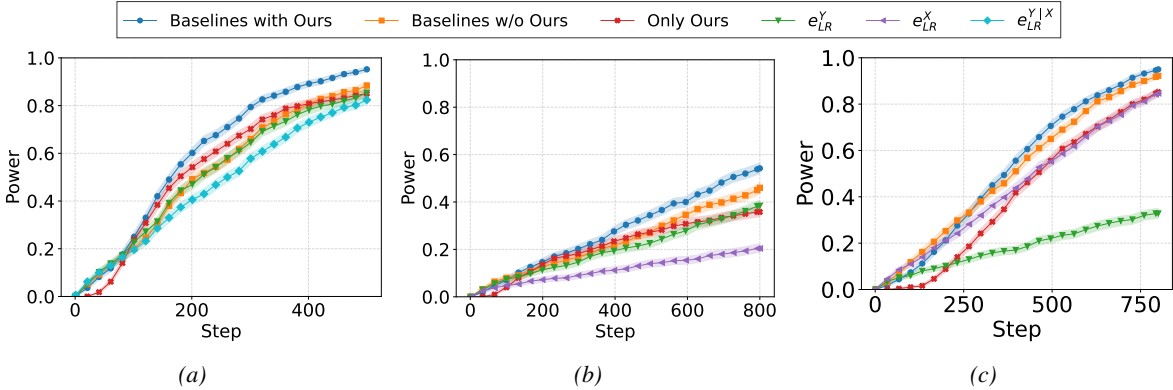

*(a)*         *(b)*         *(c)*

*Figure 3.* Power as a function of step results for online testing of improvement of a new LLM compared to an existing LLM. We use math benchmarks to evaluate the models. In all settings $Y$ is the accuracy of the model and $X$ varies between experiments: (a) $X$ is the identifier of the math question, (b) $X$ is the accuracy according to an LLM judge and we assume access to few unlabeled samples, and (c) where $X$ is the accuracy according to an LLM judge and we assume access to many unlabeled samples.

## 6.2. Concept Shift: Testing Conditional Improvement

When evaluating models using multiple datasets, our goal is to test if there is at least one dataset for which model $B$ is better than model $A$. Thus, we define $X$ to be the dataset identifier of the question. In the following experiments, we use GSM8K and AQUA-RAT datasets, so $X = 1$ if the question came from AQUA-RAT and $X = 0$ if the question came from GSM8K.

When sampling questions to evaluate the models, we use the same distribution over $X$, therefore this setting falls under the concept shift regime. We use the same configuration as in Section 5.5 with $n = 10$ and $N = 90$, and repeat the same evaluation protocol as in Section 5. Section G.1 contains a full description of the setup.

Figure 3a depicts the results. Note that the correlation between the accuracy of the model and the dataset identifier is low (0.2), but we still see that our method (red curve) gains higher power compared to the strongest baseline (green curve). In addition, the combination that includes our method (blue curve) is more powerful compared to the combination of all other baselines (orange curve).

## 6.3. Label shift: Using LLM-based Annotations

To minimize the use of human annotations, we employ LLM-judge annotations. We define $X_A$ to be the indicator of the event that model $A$'s answer is correct according to the judge. That is, $X_A = 1$ if the answer generated by model $A$ is equal to the answer generated by the judge. We define $X_B$ similarly.

We note that this scenario fits the label-shift setting as long as the accuracy of the judge is the same for outputs generated by $A$ and $B$; i.e., $X \mid Y$ is fixed under the null and alternative. We use the same configuration as in Section 5.4

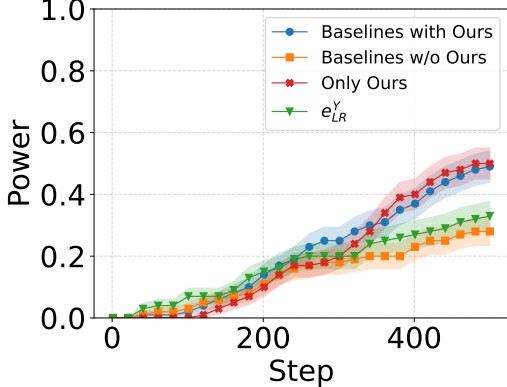

*Figure 4.* Power as a function of step results for online testing of increase in the procurement of private health insurance.

with $n = 10$ and $N \in \{10, 60\}$. We simulate each test for 800 steps and repeat the same evaluation protocol as in Section 5. We refer the reader to Section G.1 for a description of the parameters we used.

The results appear in Figures 3b and 3c. When only a few unlabeled points are available (Figure 3b), our approach contributes to the power of the combined test. Note that this setting is challenging, as evidenced by the low power obtained by all tests. In contrast, when the number of unlabeled points $N$ increases (Figure 3c), our imputed e-process diminishes but remains significant.

## 7. In-Context Learning with Multi Dimensional Mixed Type Data

Next, we evaluate our method in a more realistic setting where (i) predictions are obtained using a foundation model, and (ii) the covariates $X$ are multi-dimensional, comprising a mix of continuous and multiclass features.

We use a pretrained foundation model and provide it with the labeled data directly within its context window. We then then use it to generate predictions for a new batch of unlabeled samples. In this way, our method can benefit from the expressive power of foundation models while maintaining a low computational overhead. Indeed, naively training a foundation model at each step is practically infeasible.

To asses this approach, we consider the problem of evaluating the procurement of private health insurance using California census data (2017–2019). We define the covariate $X$ as a feature vector containing a mix of continuous and multiclass variables (e.g. sex, nativity, income, see full details at Appendix H). The target variable $Y$ is a binary indicator of private healthcare coverage. We test the hypothesis that the prevalence of private health care increased between 2017 and 2019. Our analysis shows that the distribution of $X|Y$ is fixed between those years, thus satisfying the label shift assumption. We compare our method to $e_{\text{LR}}^Y$ and $e_{\text{PPI}}$. Note that since $X$ is mixed-typed, multi-dimensional variable, modeling its distribution is highly non-trivial. As such, we do not compare to $e_{\text{LR}}^X$. As before, we plot the power of the convex combination of all baselines $\text{conv}(e_{\text{LR}}^Y, e_{\text{PPI}})$ and the power of the convex combination of the baselines with our imputed e-process: $\text{conv}(e_{\text{LR}}^Y, e_{\text{PPI}}, \breve{e}_t)$.

At each step, we draw $n = 50$ labeled points and $N = 100$ unlabeled points. Predictions are obtained with TabPFN (Hollmann et al., 2023), a state-of-the-art tabular foundation model. We repeat the experiment using 100 different realizations of the data. By pairing in-context inference with a small number of null datasets, $M = 2$, we leverage the high expressivity of a foundation model while maintaining a strictly low computational overhead.

The results are depicted in Figure 4. As can be seen, our method outperforms PPI and the likelihood ratio test (LRT). Furthermore, when combining our method with PPI and LRT we gain a significant increase in power compared to the combination of PPI and LRT.

## 8. Conclusions and Future Work

We introduced a sequential hypothesis testing procedure that leverages predictions on unlabeled data to enhance the power of statistical tests. Designed for modern settings where labeled data are scarce but covariates' data are abundant, we developed an imputed e-statistic that incorporates signals from potentially inaccurate predictions. We discuss identifiability conditions and establish guarantees for anytime finite-sample type-I error control. Focusing on binary data, we proved a non-trivial power property of our construction. Through empirical evaluation, we demonstrated the power gains over sequential LR tests and the PPI approach.

One limitation of our method is the requirement to fit the model $\mathcal{A}$ on each batch of labeled data. However, this is feasible for binary data or in situations that involve incontext learning. Another limitation is the need to sample from the null. To alleviate this, we provided a robustness result that was also validated empirically.

A natural future direction is to analyze the growth rate and expected stopping time of the method, which does not follow directly from our non-trivial power guarantees. Beyond its theoretical interest, such an analysis would yield practical guidance for choosing $\mathcal{A}$, $M$, and $K(\cdot)$ in real applications.

## Impact Statement

This paper presents work whose goal is to advance the field of machine learning and statistics. There are many potential societal consequences of our work, none of which we feel must be specifically highlighted here.

## Acknowledgments

This research was supported the European Union (ERC, SafetyBounds, 101163414). Views and opinions expressed are however those of the authors only and do not necessarily reflect those of the European Union or the European Research Council Executive Agency. Neither the European Union nor the granting authority can be held responsible for them.

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

## A. Background

### A.1. First-order stochastic dominance (FOSD).

The notion of stochastic order and FOSD in particular are at the core of this work. Formally, let $X$ and $Y$ be real-valued random variables with cumulative distribution functions $F_X(t) = \mathbb{P}(X \leq t)$ and $F_Y(t) = \mathbb{P}(Y \leq t)$. We say that $Y$ *first-order stochastically dominates* $X$ (denoted $X \leq_{\text{FOSD}} Y$) if

$$F_X(t) \; \geq \; F_Y(t) \qquad \text{for all } t \in \mathbb{R}.$$

A *necessary and sufficient* for FOSD. Equivalently, $X \leq_{\text{FOSD}} Y$ if and only if

$$\mathbb{E}[\varphi(X)] \; \leq \; \mathbb{E}[\varphi(Y)]$$

for every bounded nondecreasing function $\varphi : \mathbb{R} \to \mathbb{R}$ (Shaked & Shanthikumar, 2007).

**Strict FOSD and equivalent characterizations.** We say that $Y$ *strictly* first-order stochastically dominates $X$, denoted $X <_{\text{FOSD}} Y$, if $F_X(t) \geq F_Y(t)$ for all $t$ and the inequality is strict for at least one $t$ (equivalently, on a set of positive measure). In this case,

$$\mathbb{E}[\varphi(X)] \; < \; \mathbb{E}[\varphi(Y)]$$

for every strictly increasing function $\varphi$ for which the expectations exist.

Many members of the exponential family of distributions are FOSD ordered. Popular examples include: the Bernoulli and Binomial (that are ordered with respect to the probability of success parameter); the Poisson distribution (that is ordered with respect to the rate parameter); the exponential distribution, and the normal distribution with a fixed variance term (that is ordered with respect to its location parameter).

### A.2. Soft-rank E-value

A core concept we use is that of a *soft rank* e-value (Ramdas & Wang, 2025). Similarly to many modern testing procedures, it relies on the assumption that one can draw $M$ exchangeable samples from the null. These exchangeable samples are used to estimate the distribution of the test statistic under the null, and the null is rejected if the observed statistic is larger than the $(1 - \alpha)$ quantile of the empirical distribution.

Formally, consider testing some hypothesis $\mathcal{P}$ and let $L_0$ be the test statistic calculated from the original data. Further, let $L_1, \ldots, L_M$, be the test statistics computed with the additional $M$ null samples. Note that under the null, they are exchangeable together with $L_0$. Recall that $L_0, \ldots, L_M$ are called exchangeable if their joint distribution equals that of $(L_{\sigma(0)}, \ldots, L_{\sigma(M)})$ for any permutation $\sigma$ of $\{0, \ldots, M\}$.

To build a soft-rank e-value, we first transform $L_0, L_1, \ldots, L_B$ into some nonnegative scores $R_0, R_1, \ldots, R_B$ while preserving their order and exchangeability. Concretely,

$$E = \frac{(M + 1)R_0}{\sum_{i=0}^{M} R_i}, \tag{8}$$

where by convention $E = 1$ if $\sum_{i=1}^{M+1} R_i = 0$. It is easy to see that under the null $E$ is a valid e-value. As its name suggests, it is closely related to the standard rank-based p-value defined as $P = \frac{\sum_{i=1}^{M} \mathbb{1}_{\{L_0 \geq L_i\}}}{M+1}$. Since $R_i$ preserve ordering

$$P = \frac{\sum_{i=1}^{M} \mathbb{1}_{\{L_0 \geq L_i\}}}{M + 1} \leq \frac{\sum_{i=0}^{M} R_i / R_0}{M + 1} = \frac{1}{E}.$$

Since $P$ quantifies the rank of $L_0$ amongst $L_0, \ldots, L_B$, $1/E$ can be seen as a smoothed notion of rank, or "soft-rank".

## B. Proofs

### B.1. Proofs for Subsection 3.1

**Proof of Proposition 3.1**

**Proposition B.1.** *Let* $(D^1, \tilde{\underline{X}}^1) \ldots, (D^M, \tilde{\underline{X}}^M)$ *be drawn independently from the null distribution, and assume* $(D^i, \tilde{\underline{X}}^i)$, $1 \leq i \leq M$ *are independent of the observed* $(D, \tilde{\underline{X}})$. *Then under label shift (concept shift) setting,* $\breve{e}[D, \tilde{\underline{X}}]$ *is a valid e-statistic with respect to* $\mathcal{P}_1(\mathcal{P}_2)$: $\mathbb{E}_{H_0}[\breve{e}[D, \tilde{\underline{X}}]] = 1$, *where the expectation is taken with respect to the data* $(D, \tilde{\underline{X}})$ *and* $(D^1, \tilde{\underline{X}}^1) \ldots, (D^M, \tilde{\underline{X}}^M)$.

*Proof.* Define $(D^0, \tilde{\underline{X}}^0) = (D, \tilde{\underline{X}})$. We first show that $\breve{e}[D, \tilde{\underline{X}}]$ is a valid e-statistic for exchangeability: if $(D^0, \tilde{\underline{X}}^0), (D^1, \tilde{\underline{X}}^1) \ldots, (D^M, \tilde{\underline{X}}^M)$ are exchangeable, then:

$$\breve{e}^j[D^j, \tilde{\underline{X}}^j] = \frac{(M+1)K(\mathcal{A}[D^j, \tilde{\underline{X}}^j])}{\left(K(\mathcal{A}[D^0, \tilde{\underline{X}}^0]) + \sum_{i=1}^M K(\mathcal{A}[D^i, \tilde{\underline{X}}^i])\right)},$$

$$j = 0, \ldots, M$$

have the same expected value as $\breve{e}[D, \tilde{\underline{X}}]$. Since $\sum_j \breve{e}^j[D^j, \underline{X}^j] = M + 1$, this implies $\mathbb{E}[\breve{e}[D, \tilde{\underline{X}}]] = 1$.

Next, under the label-shift setting and given that $(D^1, \tilde{\underline{X}}^1) \ldots, (D^M, \tilde{\underline{X}}^M)$ are i.i.d and $(D^0, \tilde{\underline{X}}^0)$ is independent of $(D^1, \tilde{\underline{X}}^1) \ldots, (D^M, \tilde{\underline{X}}^M)$, we have that if $P_Y^{\text{null}} = P_Y$ then $(D^0, \tilde{\underline{X}}^0) \stackrel{d}{=} (D^i, \tilde{\underline{X}}^i)$ for $i \in \{1, \ldots, M\}$. Therefore under the label-shift setting testing $H_0(\mathcal{P}_0)$ is equivalent to testing exchangeability.

Following a similar argument we get that under concept shift, testing $H_0(\mathcal{P}_\in)$ is equivalent to testing exchangeability, which concludes the proof.

$\square$

## B.2. Proofs for Subsection 3.3

**Proof of Proposition 3.2**

**Proposition B.2.** *Let* $(D_t^1, \tilde{\underline{X}}_t^1) \ldots, (D_t^M, \tilde{\underline{X}}_t^M)$ *be drawn independently from the null distribution, and assume* $(D_t^i, \tilde{\underline{X}}_t^i)$ *are independent of the observed* $(D_t, \tilde{\underline{X}}_t), \forall 1 \leq i \leq M$. *Then under label shift (concept shift) setting,* $\breve{e}[D_t, \tilde{\underline{X}}_t]$ *is a valid conditional e-statistic with respect to* $\mathcal{P}_1(\mathcal{P}_2)$ : $\mathbb{E}_{H_0}[\breve{e}[D_t, \tilde{\underline{X}}_t] \mid \mathcal{F}_{t-1}] = 1$, *where the expectation is taken with respect to the fresh data* $(D_t, \tilde{\underline{X}}_t)$ *and* $(D_t^1, \tilde{\underline{X}}_t^1) \ldots, (D_t^M, \tilde{\underline{X}}_t^M)$.

*Proof.* As before, we will show that $\breve{e}[D_t, \tilde{\underline{X}}_t]$ is a valid conditional e-statistic for testing exchangability, and the validity for testing $H_0(\mathcal{P}_1)$ and $H_0(\mathcal{P}_2)$ will immediately follow. First, observe that given the historical data $D_{-t}$, $K_t$ is fixed. Next define $(D_t^0, \tilde{\underline{X}}_t^0) = (D_t, \tilde{\underline{X}}_t)$. Note that under the null, $(D_t^0, \tilde{\underline{X}}_t^0), (D_t^1, \tilde{\underline{X}}_t^1) \ldots, (D_t^M, \tilde{\underline{X}}_t^M)$ are exchangeable. Therefore

$$\breve{e}^j[D_t^j, X_t^j] = \frac{(M+1)K(\mathcal{A}[D_t^j, \tilde{\underline{X}}_t^j])}{\left(K(\mathcal{A}[D_t^0, \tilde{\underline{X}}_t^0]) + \sum_{i=1}^M K(\mathcal{A}[D_t^i, \tilde{\underline{X}}_t^i])\right)},$$

$$j = 0, \ldots, M$$

all have the same expected value as $\breve{e}[D_t, \tilde{\underline{X}}_t]$.

Since $\sum_j \breve{e}^j[D_t^j, \tilde{\underline{X}}_t^j] = M + 1$, this implies $\mathbb{E}[\breve{e}[D_t, \tilde{\underline{X}}_t]] = 1$.

Thus, $\breve{e}[D_t, \tilde{\underline{X}}_t]$ is a valid conditional e-statistic and for testing exchangability and under the labels-shift (concept-shift) it is valid for testing $H_0(\mathcal{P}_1)$ and $H_0(\mathcal{P}_2)$. $\square$

**Proof of Theorem 3.4:**

*Proof.* Fix $T \in \mathbb{N}$ and $\alpha \in (0, 1)$, and define the process $\hat{S}_t(\{(D_1, \tilde{\underline{X}}_1), \ldots, (D_t, \tilde{\underline{X}}_t)\}) = \prod_{i=1}^t \hat{e}^i[D_i, \tilde{\underline{X}}_i]$, $1 \leq t \leq T$. For $t > T$, set $\hat{e}^t[D_t, X_t] = 1$, so that $\hat{S}_t = \hat{S}_T$ for $t > T$. The condition $K(\cdot) > 0$ ensures that this e-value is well-defined. If $\left((D_1, \tilde{\underline{X}}_1), \ldots, (D_t, \tilde{\underline{X}}_t)\right)$ has distribution $\hat{L}_\otimes^{\text{null}}$, then the process $(\hat{S}_t)_{t \in \mathbb{N}}$ is a non-negative martingale with respect to

the filtration $\mathcal{F}_t = \sigma\left((D_1\underline{\tilde{X}}_1), \ldots, (D_t, \underline{\tilde{X}}_t)\right)$, because

$$
\begin{aligned}
\mathbb{E}[\hat{S}_t \mid (D_1, \tilde{X}_1), \ldots, (D_{t-1}, \tilde{X}_{t-1})] &= \hat{S}_{t-1}\mathbb{E}[\hat{e}^t[D_t, \underline{\tilde{X}}_t]] \\
&= \hat{S}_{t-1}\int_{D_t, \tilde{X}_t} \frac{K_t(\mathcal{A}[D_t, \tilde{X}_t])}{\int K_t(\mathcal{A}[D_t, \tilde{X}_t])d\hat{P}_t^{\text{null}}(D, \underline{\tilde{X}})}d\hat{P}_t^{\text{null}}(D, \underline{\tilde{X}}) \\
&= \hat{S}_{t-1}.
\end{aligned}
$$

Consider the event:

$$
\mathcal{C} = \{(D_1, \tilde{x}_1), \ldots, (D_t, \tilde{x}_t) : \ \exists t \leq T : \ \hat{S}_t((D_1, \tilde{x}_1), \ldots, (D_t, \tilde{x}_t)) \geq \frac{1}{\alpha}\}.
$$

The type-I error is defined as: $P_{(D_1, \underline{X}_1)\otimes\ldots\otimes(D_T, \underline{X}_T)\sim P_{\otimes_T}^{\text{null}}}(\exists t \leq T : \hat{S}_t \geq \frac{1}{\alpha}) = P_{\otimes_T}^{\text{null}}(\mathcal{C})$.

By the definition of TV-distance, we have that:

$$
P_{\otimes_T}^{\text{null}}(\mathcal{C}) \leq \hat{P}_{\otimes_T}^{\text{null}}(\mathcal{C}) + d_{TV}(P_{\otimes_T}^{\text{null}}, \hat{P}_{\otimes_T}^{\text{null}})
$$

By Ville's inequality,

$$
\hat{P}_{\otimes_T}^{\text{null}}(\mathcal{C}) = P_{(D_1, \underline{X}_1)\otimes\ldots\otimes(D_T, \underline{X}_T)\sim \hat{L}_{\otimes_T}^{\text{null}}}(\exists t \leq T : \ \hat{S}_t \geq \frac{1}{\alpha}) \leq \alpha.
$$

Therefore

$$
P_{(D_1, \tilde{X}_1)\otimes\ldots\otimes(D_T, \tilde{X}_T)\sim P_{\otimes_T}^{\text{null}}}(\exists t \leq T : S_t \geq \frac{1}{\alpha}) \leq \alpha + d_{TV}(P_{\otimes_T}^{\text{null}}, \hat{P}_{\otimes_T}^{\text{null}}),
$$

as required. $\qquad\qquad\square$

### B.3. Non Trivial Bound for TV Distance

**Theorem B.3.** *Let $(D_1, \tilde{X}_1) \stackrel{d}{=} \ldots \stackrel{d}{=} (D_t, \tilde{X}_t)$ be independent identically distributed copies of $(D, \tilde{X})$. We let $P_{\otimes_t}^{\text{null}}$ be their joint distribution under the null: $P_{\otimes_t}^{\text{null}}((D_1, \tilde{X}_1), \ldots, (D_t, \tilde{X}_t)) = \prod_{i=1}^t P_i^{\text{null}}(D_i, \tilde{X}_i)$. Assume that at every step $t$, we can sample $M_1$ labeled samples and $M_2$ unlabeled samples from the null. Let $\hat{P}_t^{\text{null}}(D, \underline{\tilde{X}})$ be the approximated distribution that is constructed using $tM_1$ samples of $(X, Y)$ and $tM_2$ samples of $X$. We define $\hat{P}_{\otimes_t}^{\text{null}}$ analogously to $P_{\otimes_t}^{\text{null}}$. Then, for any $\delta \in (0, 1)$, with probability at least $1 - \delta$, the total variation distance satisfies:*

$$
d_{TV}(P_{\otimes_t}^{\text{null}}, \hat{P}_{\otimes_t}^{\text{null}}) \leq 2\sqrt{t}\left(\sqrt{\frac{n^2 K_{XY}}{2M_1}} + \sqrt{\frac{N^2 K_X}{2M_2}}\right)
$$

*Where the confidence constants are:*

$$
\begin{aligned}
K_{XY} &= 4\ln 2 + \ln(2t/\delta) \\
K_X &= 2\ln 2 + \ln(2t/\delta)
\end{aligned}
$$

*Proof.* The proof proceeds in three steps: (1) decomposing the total variation distance of the sequence into a sum of step-wise errors, (2) bounding the parameter estimation error at each step using concentration inequalities, and (3) summing these errors over the full time horizon.

**Step 1: Decomposition of Joint Total Variation**

We wish to bound $d_{TV}(P_{\otimes_t}^{\text{null}}, \hat{P}_{\otimes_t}^{\text{null}})$. Under the true null and the estimated null, the steps are independent: $P_{\otimes_t}^{\text{null}} = \prod_{i=1}^{t} P^{\text{null}}$ and $\hat{P}_{\otimes_t}^{\text{null}} = \prod_{i=1}^{t} \hat{P}_i^{\text{null}}$. Thus, using the sub-additivity property of Total Variation distance for sequential distributions, we have:

$$d_{TV}(P_{\otimes_t}^{\text{null}}, \hat{P}_{\otimes_t}^{\text{null}}) \leq \sum_{i=1}^{t} d_{TV}\left(P^{\text{null}}(D_i, \underline{\tilde{X}}_i), \hat{P}^{\text{null}}(D_i, \underline{\tilde{X}}_i)\right).$$

At step $i$, the dataset consists of $n$ i.i.d. pairs drawn from $P_{XY}^{\text{null}}$ and $N$ i.i.d. samples drawn from $P_X^{\text{null}}$. As before, by the sub-additivity of TV for product measures, the distance for the single step $i$ is bounded by:

$$d_{TV}\left(P^{\text{null}}, \hat{P}^{\text{null}}\right) \leq n \cdot d_{TV}(P_{XY}^{\text{null}}, \hat{P}_{XY,i}^{\text{null}}) + N \cdot d_{TV}(P_X^{\text{null}}, \hat{P}_{X,i}^{\text{null}}),$$

where $\hat{P}_{XY,i}^{\text{null}}$ and $\hat{P}_{X,i}^{\text{null}}$ represents the empirical parameters estimated from the data at step $i$.

**Step 2: Concentration of Parameter Estimates**

Next, we bound the distance between the true parameters $P_{XY}^{\text{null}}$ and the empirical estimate $\hat{P}_{XY,i}^{\text{null}}$ derived from $m$ samples. For a discrete distribution over $k$ categories, Weissman's inequality states:

$$P\left(d_{TV}(P_{XY}^{\text{null}}, \hat{P}_{XY,i}^{\text{null}}) \geq \epsilon\right) \leq (2^k - 2)e^{-2m\epsilon^2} \leq 2^k e^{-2m\epsilon^2}$$

We apply this to our two estimators at every step $i \in \{1, \ldots, t\}$.

At step $i$, the estimators $\hat{P}_{XY}^{\text{null}}$ and $\hat{P}_X^{\text{null}}$ are built from: $m_{XY,i} = i \cdot M_1$ and $m_{X,i} = i \cdot M_2$ respectively.

We require the bound to hold for all $t$ steps and for both estimators ($XY$ and $X$) simultaneously. There are $2t$ total events. To ensure a global failure probability of at most $\delta$, we set the per-event failure probability to $\delta' = \frac{\delta}{2t}$.

Solving for $\epsilon$ in Weissman's inequality ($2^k e^{-2m\epsilon^2} = \delta'$):

$$\epsilon \leq \sqrt{\frac{k \ln 2 + \ln(1/\delta')}{2m}}$$

Substituting our specific sample sizes and alphabet sizes ($k = 4$ for $XY$, $k = 2$ for $X$):

$$\epsilon_{XY,i} \leq \sqrt{\frac{4 \ln 2 + \ln(2t/\delta)}{2iM_1}} \quad \text{and} \quad \epsilon_{X,i} \leq \sqrt{\frac{2 \ln 2 + \ln(2t/\delta)}{2iM_2}}$$

Let $K_{XY} = 4 \ln 2 + \ln(2t/\delta)$ and $K_X = 2 \ln 2 + \ln(2t/\delta)$.

**Step 3: Summation and Final Bound**

We substitute these high-probability error bounds back into the decomposition from Step 1.

$$d_{TV}(P_{\otimes_t}^{\text{null}}, \hat{P}_{\otimes_t}^{\text{null}}) \leq \sum_{i=1}^{t} \left[ n \cdot \sqrt{\frac{K_{XY}}{2iM_1}} + N \cdot \sqrt{\frac{K_X}{2iM_2}} \right]$$

Factor out the terms that do not depend on $i$:

$$d_{TV}(P_{\otimes_t}^{\text{null}}, \hat{P}_{\otimes_t}^{\text{null}}) \leq \left( n\sqrt{\frac{K_{XY}}{2M_1}} + N\sqrt{\frac{K_X}{M_2}} \right) \sum_{i=1}^{t} \frac{1}{\sqrt{i}}$$

Using the integral bound for the harmonic series $\sum_{i=1}^{t} \frac{1}{\sqrt{i}} \leq \int_0^t x^{-1/2}dx = 2\sqrt{t}$:

$$d_{TV}(P_{\otimes_t}^{\text{null}}, \hat{P}_{\otimes_t}^{\text{null}}) \leq 2\sqrt{t} \left( \sqrt{\frac{n^2 K_{XY}}{2M_1}} + \sqrt{\frac{N^2 K_X}{2M_2}} \right)$$

$\square$

## B.4. Proofs for Subsection 4.1

In this section, we will prove the following:

**Theorem B.4.** *Take $\mathcal{A}_\tau[D, \tilde{x}]$ as in (4) and assume that $P(X = 1|Y = 0) < P(X = 1|Y = 1)$. Then, under the label-shift regime, $\mathbb{E}_{H_1(\mathcal{P}_1)}[e[D, \tilde{X}]] > 1$. Furthermore, letting $(D^1, \underline{\tilde{X}}^1), \ldots, (D^M, \underline{\tilde{X}}^M)$ be M i.i.d data sets, drawn from the null, independently of $(D, \underline{\tilde{X}})$. Then under the alternative, both $\mathbb{E}_{H_1(\mathcal{P}_1)}[\check{e}[D, \underline{\tilde{X}}]] > 1$, and $\mathbb{E}_{H_1(\mathcal{P}_1)}[\check{e}[D_t, \underline{\tilde{X}}_t]] > 1$, where the expectation is taken with respect to $(D, \underline{\tilde{X}})$ and $(D^1, \underline{\tilde{X}}^1) \ldots, (D^M, \underline{\tilde{X}}^M)$.*

Recall that we take $A_\gamma[D, x] = \mathbb{1}\{\hat{P}(Y = 1 \mid X = x) > \gamma\}$, with $\hat{P}(Y = 1 \mid X = x)$ estimated as follows:

$$\hat{P}(Y = 1 \mid X = x) = \frac{\hat{P}_D(Y = 1)\hat{P}(x \mid Y = 1)}{\hat{P}_D(Y = 1)\hat{p}(x \mid Y = 1) + \hat{P}_D(Y = 0)\hat{P}(x \mid Y = 0)}.$$

In the above expression, $\hat{P}_D(Y = 1)$ denotes the empirical proportion of $P(Y = 1)$ in $D$, and all conditional probabilities $P(X \mid Y = y)$ are estimated using past null data. This is possible since under the label-shift regime, $P(X \mid Y = y)$ is fixed.

To derive the results, it would be convenient to represent $\mathcal{A}_\gamma[D, x]$ as follows. Define:

$$\hat{\tau}_0 = \frac{\hat{P}(X = 0|Y = 0)}{\hat{P}(X = 0|Y = 0) + \hat{P}(X = 0|Y = 1)}, \quad \hat{\tau}_1 = \frac{\hat{P}(X = 1|Y = 0)}{\hat{P}(X = 1|Y = 0) + \hat{P}(X = 1|Y = 1)},$$

It is easy to verify that for $x \in \{0, 1\}$ and any $\gamma \in (0, 1)$:

$$\mathcal{A}_\gamma[D, x] = \mathbb{1}\left\{\hat{P}_D(Y = 1) > \frac{\gamma\hat{\tau}_x}{\gamma\hat{\tau}_x + (1 - \gamma)(1 - \hat{\tau}_x)}\right\}.$$

Importantly, at each step t, $\frac{\gamma\hat{\tau}_x}{\gamma\hat{\tau}_x + (1-\gamma)(1-\hat{\tau}_x)}$ is constant as both $\gamma$ and $\hat{\tau}_x$ are fixed. With this representation in place, we first show that the population imputed e-statistic, $e[D, \underline{\tilde{X}}]$, is powered against the alternative. Toward this result, the following propositions will be useful:

**Proposition B.5.** *Let $\mathcal{A}$ be as defined above. Let $D_0, D_1$ be datasets that consist of n points drawn from $P^0_{XY}, P^1_{XY}$, respectively: $D_0 \sim \prod_{i=1}^n P^0_{XY}, D_1 \sim \prod_{i=1}^n P^1_{XY}$, where $P^0_{XY} = P_Y(; p_0)P(X|y), P^1_{XY} = P_Y(; p_1)P(X|y)$, with $p_0 < p_1$. Assume a non-degenerate setting (i.e., $0 < p_0 < p_1 < 1$) and $P(X = 1|Y = 0) < P(X = 1|Y = 1)$. Then, for any fixed binary vector $\tilde{x} \in \{0, 1\}^n$, under the label-shift regime*

$$\sum_{i=1}^N \mathcal{A}[D_0, \tilde{x}_i] <_{FOSD} \sum_{i=1}^N \mathcal{A}[D_1, \tilde{x}_i].$$

*Proof.* Let $\hat{P}_{D_0}(Y = 1), \hat{P}_{D_1}(Y = 1)$ be the empirical means in $D_0, D_1$, respectively. Note that for any fixed $n$, $\hat{P}_{D_0}(Y = 1) <_{FOSD} \hat{P}_{D_1}(Y = 1)$. Next, let $\hat{\tau}_0, \hat{\tau}_1 \in (0, 1)$ as above, and for non-negative constants $c_0, c_1$ define:

$$h(u; (\tau_0, \tau_1), (c_0, c_1)) = c_0\mathbb{1}\{u > \hat{\tau}_0\} + c_1\mathbb{1}\{u > \hat{\tau}_1\}.$$

Clearly, for $u \in (0, 1)$, $h(u)$ is non-decreasing, and since $\hat{\tau}_0, \hat{\tau}_1 \in (0, 1)$, then assuming $c_0 + c_1 > 0$, $h(u)$ is not constant. Therefore,

$$h(\hat{P}_{D_0}(Y = 1)) = c_0\mathbb{1}\{\hat{P}_{D_0}(Y = 1) > \hat{\tau}_0\} + c_1\mathbb{1}\{\hat{P}_{D_0}(Y = 1) > \hat{\tau}_1\}$$
$$<_{FOSD} c_0\mathbb{1}\{\hat{P}_{D_1}(Y = 1) > \hat{\tau}_0\} + c_1\mathbb{1}\{\hat{P}_{D_1}(Y = 1) > \hat{\tau}_1\}$$
$$= h(\hat{P}_{D_1}(Y = 1)).$$

Therefore,

$$c_0\mathcal{A}[D_0, 0] + c_1\mathcal{A}[D_0, 1] <_{FOSD} c_0\mathcal{A}[D_1, 0] + c_1\mathcal{A}[D_1, 1]$$

Finally note that for any fixed $\tilde{x} \in \{0,1\}^N$:

$$\sum_{i=1}^{N} \mathcal{A}[D_0, \tilde{x}_i] = N_0 \mathcal{A}[D_0, 0] + N_1 A[D_0, 1],$$

$$\sum_{i=1}^{N} \mathcal{A}[D_1, \tilde{x}_i] = N_0 \mathcal{A}[D_1, 0] + N_1 \mathcal{A}[D_1, 1],$$

where $N_0 = \#\{i, x_i = 0\}$, $N_1 = N - N_0$. Thus, we arrived at the desired result. $\qquad\square$

**Proposition B.6.** *Let $\mathcal{A}$ be as defined above. Let $D_0, D_1$ be datasets that consist of $n$ points drawn from $P_{XY}^0, P_{XY}^1$, respectively: $D_0 \sim \prod_{i=1}^{n} P_{XY}^0, D_1 \sim \prod_{i=1}^{n} P_{XY}^1$, where $P_{XY}^0 = P_Y(; p_0)P(X|y), P_{XY}^1 = P_Y(; p_1)P(X|y)$, with $p_0 < p_1$. Assume (i) a non-degenerate setting: $0 < p_0 < p_1 < 1$ and $P(X = 1|Y = 0) < P(X = 1|Y = 1)$. Let $\tilde{\underline{X}}^0 \sim \prod_{i=1}^{N} P_X^0, \tilde{\underline{X}}^1 \sim \prod_{i=1}^{N} P_X^1$. Then:*

$$\sum_{i=1}^{N} \mathcal{A}[D_0, \tilde{X}_i^0] <_{FOSD} \sum_{i=1}^{N} \mathcal{A}[D_1, \tilde{X}_i^1].$$

*Proof.* First note that for a fixed $N$:

$$\sum_{i=1}^{N} \mathcal{A}[D_0, \tilde{X}_i^0] = (N - N_{1,0})\mathcal{A}[D_0, 0] + N_{1,0}\mathcal{A}[D_0, 1],$$

where $N_{1,0} = \#\{i : \tilde{X}_i^0 = 1\}$ and $N_{1,0} \sim Bin(N, p_0)$. Similarly,

$$\sum_{i=1}^{N} \mathcal{A}[D_1, \tilde{X}_i^1] = (N - N_{1,1})\mathcal{A}[D_1, 0] + N_{1,1}\mathcal{A}[D_1, 1],$$

where $N_{1,1} = \#\{i : \tilde{X}_i^1 = 1\}$ and $N_{1,1} \sim Bin(N, p_1)$. Furthermore, let $p_0 = P(X^0 = 1), p_1 = P(X^1 = 1)$, we have

$$
\begin{aligned}
p_0 &= P(\tilde{X}^0 = 1) \\
&= P(Y^{\text{null}} = 0)P(\tilde{X}^0 = 1|Y^{\text{null}} = 0) + P(Y^{\text{null}} = 1)P(\tilde{X}^0 = 1|Y^{\text{null}} = 1) \\
&< P(Y^{\text{alt}} = 0)P(\tilde{X}^0 = 1|Y^{\text{null}} = 0) + P(Y^{\text{alt}} = 1)P(\tilde{X}^0 = 1|Y^{\text{null}} = 1) \\
&= P(Y^{\text{alt}} = 0)P(\tilde{X}^1 = 1|Y^{\text{alt}} = 0) + P(Y^{\text{alt}} = 1)P(\tilde{X}^1 = 1|Y^{\text{alt}} = 1) \\
&= P(\tilde{X}^1 = 1) \\
&= p_1,
\end{aligned}
$$

where we used the fact that under the label-shift regime $P(X = x|Y = y)$ are fixed, and the assumption that $P(X = 1|Y = 0) < P(X = 1|Y = 1)$. Therefore,

$$
\begin{aligned}
P(\sum_{i=1}^{N} \mathcal{A}[D_0, \tilde{X}_i^0] \le t) &= P((N - N_{1,0})\mathcal{A}[D_0, 0] + N_{1,0}\mathcal{A}[D_0, 1] \le t) \\
&= \sum_{k=0}^{N} P(N_{1,0} = k)P((N - k)\mathcal{A}[D_0, 0] + k\mathcal{A}[D_0, 1] \le t) \\
&\ge \sum_{k=0}^{N} P(N_{1,1} = k)P((N - k)\mathcal{A}[D_0, 0] + k\mathcal{A}[D_0, 1] \le t) \\
&\ge \sum_{k=0}^{N} P(N_{1,1} = k)P((N - k)\mathcal{A}[D_1, 0] + k\mathcal{A}[D_1, 1] \le t) \\
&= P(\sum_{i=1}^{N} \mathcal{A}[D_1, \tilde{X}_i^1] \le t).
\end{aligned}
$$

The first inequality is due to fact that the map $k \to P((N-k)\mathcal{A}[D_0,0] + k\mathcal{A}[D_0,1] \leq t)$ is non-increasing in $k$. To see this, let $S_k = (N-k)\mathcal{A}[D_0,0] + k\mathcal{A}[D_0,1]$. Then $S_{k+1} = S_k + (\mathcal{A}[D_0,1] - \mathcal{A}[D_0,0])$. Thus,

$$P(S_{k+1} \leq t) = \sum_{0 \leq m \leq t} P(S_k = m)P(\mathcal{A}[D_0,1] - \mathcal{A}[D_0,0] \leq t - m | S_k = m)$$

$$\leq \sum_{0 \leq m \leq t} P(S_k = m) = P(S_k \leq t).$$

Therefore, for any $k < k'$

$$P((N-k)\mathcal{A}[D_0,0] + k\mathcal{A}[D_0,1] \leq t) \geq P((N-k')\mathcal{A}[D_0,0] + k'\mathcal{A}[D_0,1] \leq t).$$

In words, the mixing distribution of $N_{1,1}$ assigns more weight to higher values of $k$, whose corresponding probabilities are lower or equal. The second inequality is due to the fact that for any fixed $k$

$$(N-k)A[D_0,0] + k\mathcal{A}[D_0,1] <_{FOSD} (N-k)\mathcal{A}[D_1,0] + kA[D_1,1]$$

Finally, since for each $k$, $(N-k)\mathcal{A}[D_0,0] + k\mathcal{A}[D_0,1] <_{FOSD} (N-k)\mathcal{A}[D_1,0] + k\mathcal{A}[D_1,1]$, there exists $t_0$ such that

$$p((N-k)\mathcal{A}[D_0,0] + k\mathcal{A}[D_0,1] \leq t_0) > p((N-k)\mathcal{A}[D_1,0] + k\mathcal{A}[D_1,1] \leq t_0).$$

Hence,

$$P(\sum_{i=1}^{n} \mathcal{A}[D_0, \tilde{X}_i^0] \leq t_0) > P(\sum_{i=1}^{n} \mathcal{A}[D_1, \tilde{X}_i^1] \leq t_0),$$

and the domination is strict. $\qquad \square$

As we show next, based on the last result, we can show that the population and finite sample e-statistics are powered against the alternative. We start with the population version $e[D, \underline{\tilde{X}}]$:

**Under label-shift regime, $e[D, \underline{\tilde{X}}]$ has non-trivial power against the alternative:**

*Proof.* Let $\tilde{X}_0 \sim P_X^{\text{null}}, \tilde{X}_1 \sim P_X^{\text{alt}}$. By the above proposition, $\mathcal{A}[D_0, \tilde{X}_0] <_{FOSD} \mathcal{A}[D_1, \tilde{X}_1]$. The result directly follows from the fact that $\psi(\cdot)$ is strictly increasing, since in this case $\mathbb{E}_{H_0}[\psi((\mathcal{A}[D_0, \tilde{X}_0])] < \mathbb{E}_{H_1}[\psi((\mathcal{A}[D_1, \tilde{X}_1])]$. $\qquad \square$

Next we show that the finite-sample e-statistic $\breve{e}[D, \underline{\tilde{X}}]$ is also powered against the alternative under S label-shift setting:

**Under label-shift regime, $\breve{e}[D, \underline{\tilde{X}}]$ has non-trivial power against the alternative:**

*Proof.* Let $(D^0, \underline{\tilde{X}}^0) = (D, \underline{\tilde{X}})$, and define

$$\breve{e}^j[D^j, \underline{\tilde{X}}^j] = \frac{(M+1)\sum_{k=1}^{N} \psi(\mathcal{A}[D^j, \tilde{X}_k^j])}{\left(\sum_{j=1}^{N} \psi(\mathcal{A}[D^0, \tilde{X}_j^0]) + \sum_{i=1}^{M} \sum_{j=1}^{N} \psi(\mathcal{A}[D^i, \tilde{X}_j^i])\right)},$$
$$j = 0, \ldots, M.$$

For $j \neq 0$, by law of total expectation

$$\mathbb{E}[\breve{e}^j[D^j, \underline{\tilde{X}}^j]] = \mathbb{E}_{(D^1, \underline{\tilde{X}}^1), \ldots, (D^M, \underline{\tilde{X}}^M)}[\mathbb{E}_{(D^0, \underline{\tilde{X}}^0)} \left[\breve{e}^j[D^j, \underline{\tilde{X}}^j] | (D^1, \underline{\tilde{X}}^1) \ldots, (D^M, \underline{\tilde{X}}^M)\right]$$

Next, note that for fixed $(D^1, \underline{\tilde{X}}^1), \ldots, (D^M, \underline{\tilde{X}}^M)$, the inner integrand of the RHS can be written as follows:

$$\mathbb{E}_{(D^0, \tilde{X}^0)}\left[\frac{C_0}{\sum_i \psi(\mathcal{A}[D^0, \tilde{X}_i^0]) + C_1}\right] =$$

$$\mathbb{E}_{(D^0, \tilde{X}^0)}\left[\frac{C_0}{\psi(1)\sum_i \mathcal{A}[D^0, \tilde{X}_i^0] + \psi(0)(N - \sum_i \mathcal{A}[D^0, \tilde{X}_i^0]) + C_1}\right],$$

where $C_0, C_1$ are positive constants. Since $\psi(\cdot)$ is strictly increasing, so is

$$g(s) = \psi(1)s + \psi(0)(N - s), \ \ s = 0, \ldots, N.$$

Hence

$$g^*(s) = \frac{C_0}{g(s) + C_1}$$

is strictly decreasing in s. Next, let $(D^*, \underline{X}^*) \sim P^{\text{null}}(D, \tilde{\underline{X}})$ independent of $(D^1, \tilde{\underline{X}}^1), \ldots, (D^M, \tilde{\underline{X}}^M)$.

By Proposition B.6, $\sum_{j=1}^N \mathcal{A}[D^*, X_j^*] <_{FOSD} \sum_{j=1}^N \mathcal{A}[D^0, \tilde{X}_j^0]$. Thus,

$$\mathbb{E}_{(D^0, \tilde{\underline{X}}^0)} \left[ \frac{C_0}{g(\sum_{j=1}^N \mathcal{A}[D^0, \tilde{X}_j^0]) + C_1} \right] < \mathbb{E}_{(D^*, \underline{X}^*)} \left[ \frac{C_0}{g(\sum_{j=1}^N \mathcal{A}[D^*.X_j^*]) + C_1} \right],$$

Therefore

$$\mathbb{E}[\breve{e}^j[D^j, \tilde{\underline{X}}^j]] < \mathbb{E}_{(D^1, \tilde{\underline{X}}^1), \ldots, (D^M, \tilde{\underline{X}}^M)} \mathbb{E}_{(D^*, \underline{X}^*)} \left[ \breve{e}^j[D^j, \tilde{\underline{X}}^j] \mid (D^1, \tilde{\underline{X}}^1) \ldots, (D^M, \tilde{\underline{X}}^M) \right],$$

where the expectation on the LHS is with respect to the observed data $(D^0, \tilde{X}^0)$ and the $M$ datasets $(D^1, \tilde{\underline{X}}^1) \ldots, (D^M, \tilde{\underline{X}}^M)$.

Importantly, $(D^1, \tilde{\underline{X}}^1) \ldots, (D^M, \tilde{\underline{X}}^M)$ and $(D^*, X^*)$ are all i.i.d. copies drawn from the null, and thus are exchangeable. Therefore, by Theorem 3.1

$$\mathbb{E}_{(D^1, \tilde{\underline{X}}^1), \ldots, (D^M, \tilde{\underline{X}}^M)} \mathbb{E}_{(D^*, \underline{X}^*)} \left[ \breve{e}^j[D^j, \tilde{\underline{X}}^j] \mid (D^1, \tilde{\underline{X}}^1) \ldots, (D^M, \tilde{\underline{X}}^M) \right] = 1.$$

Thus,

$$\mathbb{E}[\breve{e}^j[D^j, \tilde{\underline{X}}^j]] < 1, \ \ 1 \le j \le M$$

Finally, since $\sum_{j=0}^M \mathbb{E}[\breve{e}^j[D^j, \tilde{\underline{X}}^j]] = M + 1$, and $\{\mathbb{E}[\breve{e}^j[D^j, \tilde{\underline{X}}^j]]\}_{j=1}^M$ are all equal,

$$\mathbb{E}[\breve{e}^0[D^0, \tilde{\underline{X}}^0]] = (M + 1) - M\mathbb{E}[\breve{e}^j[D^1, \tilde{\underline{X}}^1]] > (M + 1) - M \cdot 1 = 1,$$

as required. $\qquad \square$

Finally, it remains to show that the conditional imputed e-statistic has also non-trivial power against the null,i.e., that $\mathbb{E}_{H_1(\mathcal{P}_1))}[e[D_t, \tilde{\underline{X}}] \mid \mathcal{F}_{t-1}] > 1$:

**Under label-shift regime, $\breve{e}[D_t, \tilde{\underline{X}}_t]$ has non-trivial power against the alternative:**

*Proof.* This follows directly from the results above as given the historical data $K_t$ is fixed, and we already showed that $\mathbb{E}_{H_1(\mathcal{P}_1)}[e(D, \tilde{\underline{X}})] > 1$ for any fixed positive and strictly increasing function $K$. Thus we are back to the unconditional case, and the proof is complete. $\qquad \square$

## B.5. Proofs for Subsection 4.2

In this section, we will prove the following:

**Theorem B.7.** *Take $\mathcal{A}[D, \tilde{x}]$ as in (6) and assume that $P(X = 1|Y = 0) < P(X = 1|Y = 1)$. Then, under the concept shift regime, $\mathbb{E}_{H_1(\mathcal{P}_2^+)}[e[D, \tilde{X}]] > 1$. Furthermore, letting $(D^1, \tilde{\underline{X}}^1), \ldots, (D^M, \tilde{\underline{X}}^M)$ be M i.i.d data sets, drawn from the null, independently of $(D, \tilde{\underline{X}})$. Then both $\mathbb{E}_{H_1(\mathcal{P}_2^+)}[\breve{e}[D, \tilde{\underline{X}}]] > 1$ and $\mathbb{E}_{H_1(\mathcal{P}_2^+)}[\breve{e}[D_t, \tilde{\underline{X}}_t]] > 1$, where the expectation is taken with respect to $(D, X)$ and $(D^1, \tilde{\underline{X}}^1) \ldots, (D^M, \tilde{\underline{X}}^M)$.*

To prove Theorem 4.2, the following propositions will be useful:

**Proposition B.8.** *Let $Z_0 \sim Ber(P_0)$, $Z_1 \sim Ber(\mathcal{P}_1)$, where $P_0, P_1$ are random success probabilities. Then $P_0 \le_{FOSD} \mathcal{P}_1 \Rightarrow Z_0 \le_{FOSD} Z_1$.*

*Proof.* By the law of total expectation:

$$P(Z_i = 1) = \mathbb{E}_{P_i}[Z_i | P_i = p] = \mathbb{E}_{P_i}[p], \ i \in \{0, 1\}.$$

Next, let $\psi(p)$ be any real-value non-decreasing function. Then, since $P_0 \leq_{FOSD} \mathcal{P}_1$, we have $\mathbb{E}_{P_0}[\psi(p)] \leq \mathbb{E}_{P_1}[\psi(p)]$. In particular, taking $\psi(p) = p$ we get that $P(Z_0 = 1) \leq P(Z_1 = 1)$. The last inequality then implies the desired result. $\qquad\square$

**Proposition B.9.** *Fix $n \geq 1$. For $i \in \{0, 1\}$, let $\{(Y_{i,j}, X_{i,j})\}_{j=1}^n$ be i.i.d. draws with*

$$X_{i,j} \sim \text{Bernoulli}(r), \qquad r \in (0, 1),$$

*and let*

$$\theta_i(x) := \Pr(Y_{i,j} = 1 \mid X_{i,j} = x).$$

*Assume that for every x, $\theta_0(x) \leq \theta_1(x)$ and for $N_i(x) > 0$, define $N_i(x) = \sum_{j=1}^n \mathbf{1}\{X_{i,j} = x\}$,*

$$q_i(x) = \begin{cases} \frac{1}{N_i(x)} \sum_{j=1}^n \mathbf{1}\{X_{i,j} = x, Y_{i,j} = 1\} & \text{if } N_i(x) > 0 \\ 0 & \text{if } N_i(x) = 0 \end{cases}.$$

*Then for every x, $q_0(x) \leq_{FOSD} q_1(x)$.*

*Moreover, if $\theta_0(x) < \theta_1(x)$, the dominance is strict.*

*Proof.* Fix $x \in \{0, 1\}$ and $m \in \{0, \dots, n\}$, and assume for now that $N_0(x) = N_1(x) = m$, the number of successes is then

$$K_i(x) = \sum_{j=1}^n \mathbf{1}\{X_{i,j} = x, Y_{i,j} = 1\} \sim \text{Binomial}(m, \theta_i(x)), \quad q_i(x) = \begin{cases} K_i(x)/m & m > 0 \\ 0 & m = 0 \end{cases}$$

Next, let $U_0, \dots, U_m \sim \text{Uniform}(0, 1)$ i.i.d. and define:

$$Z_{0,k} = \mathbf{1}\{U_k \leq \theta_0(x)\}, \qquad Z_{1,k} = \mathbf{1}\{U_k \leq \theta_1(x)\}.$$

Since $\theta_0(x) \leq \theta_1(x)$, we have

$$K_0(x) = \sum_{k=0}^m Z_{0,k} \leq_{FOSD} \sum_{k=0}^m Z_{1,k} = K_1(x)$$

Equivalently, since $m$ was fixed, we have that for every $t \in [0, 1]$ and all $m$,

$$\Pr(q_0(x) \leq t \mid N_0 = m) \ \geq \ \Pr(q_1(x) \leq t \mid N_1 = m). \tag{1}$$

Next, since $X_{i,j} \sim \text{Bernoulli}(r)$ with the same $r$, we have

$$N_0 \overset{d}{=} N_1 \sim \text{Binomial}(n, r).$$

Let $w_m = \Pr(N_i = m)$, independent of $i$. Therefore,

$$\Pr(q_i(x) \leq t) = \sum_{m=0}^n w_m \Pr(q_i(x) \leq t | N_i = m).$$

Applying (1) term-wise yields $q_0(x) \leq_{FOSD} q_1(x)$. Finally, if $\theta_0(x) < \theta_1(x)$, then $K_0(x) <_{FOSD} K_1(x)$, so for some $t$ the inequality is strict, and we arrive at the desired result.

$$\square$$

**Proposition B.10.** *Let $\mathcal{A}$ be the Baye's classifier. Let $D_0 \sim \prod_{i=1}^n P_{XY}^{\text{null}}$ and $D_1 \sim \prod_{i=1}^n P_{XY}^{\text{alt}}$, for some $P_{XY}^{\text{alt}} \in H_1(\mathcal{P}_2^+)$. Define $N_0^i = \#\{(x_i, y_i) \in D_i : x_i = 0\}, N_1^i = \#\{(x_i, y_i) \in D_i : x_i = 1\}$. Assume that $P_X$ is fixed, and let $\tilde{X}, \tilde{X}^* \sim P_X$, then $\mathcal{A}[D_0, \tilde{X}] <_{FOSD} \mathcal{A}[D_1, \tilde{X}^*]$.*

*Proof.* Fix $X = x$, and let $N_0(x) := |\{(x_i, y_i) \in D_0 : x_i = x\}|$. Define $N_1$ similarly. First we will show that $\mathcal{A}[D_0, x] \leq_{FOSD} \mathcal{A}[D_1, x]$. To see this, note that for a fixed $D_i$, $\mathcal{A}[D_i = D, x]$ is a Bernoulli variable. Therefore, $\mathcal{A}[D_i, x]$ is a mixture of Bernoulli variables, and thus a Bernoulli variable. Denote its probability of success by $\hat{p}_{D_i}(x)$. By Proposition B.8, it is enough to show that for a fixed $X = x$, as random variables, $\hat{p}_{D_0}(x) \leq_{FOSD} \hat{p}_{D_1}(x)$. However, under the alternative, for each $x$ : $P^{\text{null}}(Y = 1 \mid X = x) \leq P^{alt}(Y = 1 \mid X = x)$. Therefore, by Proposition B.9, indeed, $\hat{p}_{D_0}(x) \leq_{FOSD} \hat{p}_{D_1}(x)$. Furthermore, for some value of $x$, the inequality is strict. Therefore, for every value of $x$, $P(\mathcal{A}[D_0, x] = 1) \leq P(\mathcal{A}[D_1, x] = 1)$ and for some value of $x$ the inequality is strict. By law of total expectation

$$
\begin{aligned}
p_0 = P(\mathcal{A}[D_0, \tilde{X}] = 1) &= \sum_{x \in X} P(\tilde{X} = x) P(\mathcal{A}[D_0, x] = 1) \\
&< \sum_{x \in X} P(\tilde{X}^* = x) P(\mathcal{A}[D_1, x] = 1) \\
&= P(\mathcal{A}[D_1, \tilde{X}^*] = 1) = p_1,
\end{aligned}
$$

where the second inequality holds due to the fact that $P_X$ is fixed. $\qquad \square$

**Remark:** It is easy to see that Proposition B.10 holds for any $P^0_{XY}(x, y) = P^0_X(x) P^0(y|x)$ and $P^1_{XY}(x, y) = P^1_X(x) P^0(y|x)$ such that $P^0(Y = 1|x) < P^1(Y = 1|x)$ for all $x$ and $P^0_X = P^1_X = P_X$.

We are now ready to prove the that population e-statistic $e[D, \underline{\tilde{X}}]$ has non-trivial power:

**Under concept-shift regime, $e[D, \tilde{X}]$ has non-trivial power against any $Q \in H_1(\mathcal{P}_2^+)$:**

*Proof.* Let $\tilde{X}, \tilde{X}^* \sim P_X$. By the above proposition, $\mathcal{A}[D_0, \tilde{X}] <_{FOSD} \mathcal{A}[D_1, \tilde{X}^*]$. The result directly follows from the fact that $\psi(\cdot)$ is strictly increasing, thus $\mathbb{E}_{H_0(\mathcal{P}_2)}[\psi((\mathcal{A}[D_0, \tilde{X}])] < \mathbb{E}_{H_1(\mathcal{P}_2^+)}[\psi((\mathcal{A}[D_1, \tilde{X}^*])]$. $\qquad \square$

Our next goal is to prove that the finite-sample e-statistic $\breve{e}[D, \tilde{X}]$ has non-trivial power. To facilitate this result, we first generalize Proposition B.10 as follows:

**Lemma B.11.** *Let $D_0 \sim \prod_{i=1}^n P^{\text{null}}_{XY}, D_1 \sim \prod_{i=1}^n P^{\text{alt}}_{XY}$, and $X_1, \ldots, X_N$ and $\tilde{X}_1^*, \ldots, \tilde{X}_N^*$ be drawn independently from $P_X$. Let $\mathcal{A}[D_i, \tilde{X}_1], \ldots \mathcal{A}[D_i, \tilde{X}_N]$, $i \in \{0, 1\}$, be as defined above. Then $\sum_{j=1}^N \mathcal{A}[D_0, \tilde{X}_j] < \sum_{j=1}^N \mathcal{A}[D_1, \tilde{X}_j^*]$ in the FOSD order.*

*Proof.* First note that given $D_0$, $\mathcal{A}[D_0, \tilde{X}]$ is a Bernoulli variable, with probability of success $\hat{p}_0 = P(X = 0)\hat{P}(Y = 1|X = 0) + P(X = 1)\hat{P}(Y = 1|X = 1)$. Furthermore, given $D_0$, $\mathcal{A}[D_0, \tilde{X}_i], \mathcal{A}[D_0, \tilde{X}_j]$ are independent, for any $i \neq j$. Therefore, given $D_0$, $\sum_{j=1}^N \mathcal{A}[D_0, \tilde{X}_j]$ is a binomial variable with probability of success $\hat{p}_0$, and N trials. Similarly, $\sum_{j=1}^N \mathcal{A}[D_1, \tilde{X}_j^*]$ is a binomial variable with probability of success $\hat{p}_1$, and N trials. Hence, $\sum_{j=1}^N \mathcal{A}[D_0, \tilde{X}_j], \sum_{j=1}^N \mathcal{A}[D_1, \tilde{X}_j^*]$ are mixtures of Binomial variables. By Proposition B.8, it is enough to show that, as random variables, $\hat{p}_0 <_{FOSD} \hat{p}_1$. Next, observe that

$$
\hat{p}_0 = P(\tilde{X} = 0) \frac{\sum_j \mathbb{1}\{X_j^0 = 0, Y_j^{\text{null}} = 1\}}{N_0^0} + P(\tilde{X} = 1) \frac{\sum_j \mathbb{1}\{X_j^0 = 1, Y_j^{\text{null}} = 1\}}{N_1^0}.
$$

Similarly,

$$
\hat{p}_1 = P(\tilde{X}^* = 0) \frac{\sum_j \mathbb{1}\{X_j^1 = 0, Y_j^{\text{alt}} = 1\}}{N_0^1} + P(\tilde{X}^* = 1) \frac{\sum_j \mathbb{1}\{X_j^1 = 1, Y_j^{\text{alt}} = 1\}}{N_1^1},
$$

where $N_0^i = \#\{j : X_j^i = 0\}, N_1^i = \#\{j : X_j^i = 1\}$, $i \in \{0, 1\}$. Fixing $D_x^0 = D_x^1 = D_x$, $N_0^i, N_1^i$ are all constants, and all indicators in the numerators are independent. Furthermore, given $D_x$, the indicators are coupled and for each j such that $x_j = 0$: $\mathbb{1}\{x_j = 0, Y_j^{\text{null}} = 1\} \leq_{FOSD} \mathbb{1}\{x_j = 0, Y_j^{\text{alt}} = 1\}$. Similarly, for each j such that $x_j = 1$: $\mathbb{1}\{x_j = 1, Y_j^{\text{null}} = 1\} \leq_{FOSD} \mathbb{1}\{x_j = 1, Y_j^{\text{alt}} = 1\}$. Moreover, by assumption, for either $x = 0$ or $x = 1$, the dominance is strict. Thus, letting $\psi(\cdot)$ be a non-decreasing function:

$$
\mathbb{E}[\psi(\hat{p}_0)|D_x] \leq \mathbb{E}[\psi(\hat{p}_1)|D_x].
$$

Lastly, since $D_x^0 \overset{d}{=} D_x^1$, we have that

$$\mathbb{E}[\psi(\hat{p}_0)] = \mathbb{E}_{D_x}\mathbb{E}[\psi(\hat{p}_0)|D_x]] \leq \mathbb{E}_{D_x}\mathbb{E}[\psi(\hat{p}_1)|D_x] = \mathbb{E}[\psi(\hat{p}_1)].$$

Since $\psi(\cdot)$ is arbitrary non-decreasing function, we arrive at the desired result. $\qquad\square$

**Remark:** It is easy to see that Proposition B.11 holds for any $P_{XY}^0(x, y) = P_X^0(x)P^0(y|x)$ and $P_{XY}^1(x, y) = P_X^1(x)P^0(y|x)$ such that $P^0(Y = 1|x) < P^1(Y = 1|x)$ for all $x$ and $P_X^0 = P_X^1 = P_X$.

We are now ready to prove that under the concept-shift regime, $\breve{e}[D, \underline{\tilde{X}}]$ has non-trivial power.

**Under concept-shift $\breve{e}[D, \underline{\tilde{X}}]$ has non-trivial power against any $Q \in H_1(\mathcal{P}_2^+)$:**

*Proof.* The proof follows the same arguments as under the label-shift case above. $\qquad\square$

Finally, it remains to show that the conditional imputed e-statistic has also non-trivial power against the null, i.e., that $\mathbb{E}_{H_1(\mathcal{P}_2^+))}[e[D_t, \underline{\tilde{X}}_t] \mid \mathcal{F}_{t-1]}] > 1$:

**Under concept-shift regime, $\breve{e}[D_t, \underline{\tilde{X}}_t]$ has non-trivial power against any $Q \in H_1(\mathcal{P}_2^+)$:**

*Proof.* This follows directly from the results above as given the historical data $K_t$ is fixed, and we already showed that $\mathbb{E}_{H_1(\mathcal{P}_2^+)}[e(D, \underline{\tilde{X}})] > 1$ for any fixed positive and strictly increasing function $K$. Thus, we are back to the unconditional case, and the proof is complete. $\qquad\square$

## C. Extension to Composite Null under Label-Shift

In this section we wish to show that in the binary case, $\mathcal{P}_1$ can be extended to composite null testing:

$$(\mathcal{P}_1^{\text{composite}}) \quad H_0: \ \theta_Y \leq \theta_Y^{\text{null}} \quad \text{versus} \quad H_1: \ \theta_Y > \theta_Y^{\text{null}}.$$

Recall that when hypothesizing about $P_Y$, for identifiability, we assume a label-shift setting, i.e, $P(X|y)$ is fixed for all $y$. Assume $K$ is of the form $K(\underline{y}) = \sum_{i=1}^N \psi(\tilde{y}_i)$, where $\psi(\cdot)$ is a positive increasing function (e.g. $\psi(y_i) = \exp(\gamma \tilde{y}_i)$).

We first update the definition of the population imputed e-statistic to match the composite null setting as follows:

$$e[D, \underline{\tilde{X}}] = \frac{K(\underline{\tilde{Y}})}{\mathbb{E}_{P^{\text{null}}(D, \underline{\tilde{X}})}[K(\underline{\tilde{Y}})]},$$

where $P_{XY}^{\text{null}} = P_Y(; \theta_Y^{\text{null}})P(X|y)$. Let $(D, \underline{\tilde{X}}) \sim P = P_Y(; \theta_Y)P(X|y) \in H_0(\mathcal{P}_1^{\text{composite}})$. We begin by showing that the population imputed e-statistic, $e[D, \underline{\tilde{X}}]$, is valid when testing a composite null.

**Under label-shift regime, $e[D, \underline{\tilde{X}}]$ is valid when testing a composite null**

*Proof.* let $(D^*, \underline{\tilde{X}}^*) \sim P_{XY}^{\text{null}}$, $(D, \underline{\tilde{X}}) \sim P \in H_0(\mathcal{P}_1^{\text{composite}})$, and observe that by Theorem B.6,

$$\sum_{i=1}^N \mathcal{A}[D, \tilde{X}_i] <_{FOSD} \sum_{i=1}^N \mathcal{A}[D^*, \tilde{X}_i^*].$$

Since $K$ is strictly increasing, we then have

$$\mathbb{E}_{P(D, \underline{\tilde{X}})}[K(\underline{\tilde{Y}})] \leq \mathbb{E}_{P^{\text{null}}(D^*, \underline{\tilde{X}})}[K(\underline{\tilde{Y}})].$$

Thus, $\mathbb{E}_P\left[\frac{K(\underline{\tilde{Y}})}{\mathbb{E}_{P^{\text{null}}(D, \underline{\tilde{X}})}[K(\underline{\tilde{Y}})]}\right] \leq 1$, as needed. $\qquad\square$

Next we turn to establish the validity of the finite-sample imputed e-statistic in the composite null setting:

**Under label-shift regime, $\breve{e}[D, \tilde{X}]$ is valid when testing a composite null**

*Proof.* Let $(D^0, \tilde{X}^0) = (D, \tilde{X})$ be the observed data, and let $(D^1, \tilde{X}^1), \ldots, (D^M, \tilde{X}^M)$ be $M$ i.i.d datasets drawn from $P^{\text{null}}$.

Define

$$\breve{e}^j[D^j, \tilde{X}^j] = \frac{(M+1)\sum_{k=1}^N \psi(\mathcal{A}[D^j, \tilde{X}_k^j])}{\left(\sum_{j=1}^N \psi(\mathcal{A}[D^0, \tilde{X}_j^0]) + \sum_{i=1}^M \sum_{j=1}^N \psi(\mathcal{A}[D^i, \tilde{X}_j^i])\right)},$$
$$j = 0, \ldots, M.$$

For $j \neq 0$, by the law of total expectation

$$\mathbb{E}[\breve{e}^j[D^j, \tilde{X}^j]] = \mathbb{E}_{(D^1, \tilde{X}^1), \ldots, (D^M, \tilde{X}^M)}[\mathbb{E}_{(D^0, \tilde{X}^0)}\left[\breve{e}^j[D^j, \tilde{X}^j] | (D^1, \tilde{X}^1) \ldots, (D^M, \tilde{X}^M)\right]$$

Next, note that for fixed $(D^1, \tilde{X}^1), \ldots, (D^M, \tilde{X}^M)$, the inner integrand of the RHS can be written as follows:

$$\mathbb{E}_{(D^0, \tilde{X}^0)}\left[\frac{C_0}{\sum_i \psi(\mathcal{A}[D^0, \tilde{X}_i^0]) + C_1}\right] =$$
$$\mathbb{E}_{(D^0, \tilde{X}^0)}\left[\frac{C_0}{\psi(1)\sum_i \mathcal{A}[D^0, \tilde{X}_i^0] + \psi(0)(N - \sum_i \mathcal{A}[D^0, \tilde{X}_i^0]) + C_1}\right],$$

where $C_0, C_1$ are positive constants. Since $\psi(\cdot)$ is strictly increasing, so is

$$g(s) = \psi(1)s + \psi(0)(N - s), \quad s = 0, \ldots, N.$$

Hence

$$g^*(s) = \frac{C_0}{g(s) + C_1}$$

is strictly decreasing in s. Next, let $(D^*, \tilde{X}^*) \sim P^{\text{null}}$ independent of $(D^1, \tilde{X}^1), \ldots, (D^M, \tilde{X}^M)$.

By Proposition B.6, $\sum_{j=1}^N \mathcal{A}[D^0, \tilde{X}_j^0] <_{FOSD} \sum_{j=1}^N \mathcal{A}[D^*, \tilde{X}_j^*]$. Thus,

$$\mathbb{E}_{(D^*, \tilde{X}^*)}\left[\frac{C_0}{g(\sum_{j=1}^N \mathcal{A}[D^*, \tilde{X}_j^*]) + C_1}\right] < \mathbb{E}_{(D^0, X^0)}\left[\frac{C_0}{g(\sum_{j=1}^N \mathcal{A}[D^0, X_j^0]) + C_1}\right],$$

Therefore

$$\mathbb{E}[\breve{e}^j[D^j, \tilde{X}^j]] > \mathbb{E}_{(D^1, \tilde{X}^1), \ldots, (D^M, \tilde{X}^M)}\mathbb{E}_{(D^*, \tilde{X}^*)}\left[\breve{e}^j[D^j, \tilde{X}^j] \mid (D^1, \tilde{X}^1) \ldots, (D^M, \tilde{X}^M)]\right],$$

where the expectation on the LHS is with respect to the observed data $(D^0, \tilde{X}^0)$ and the $M$ datasets $(D^1, \tilde{X}^1) \ldots, (D^M, \tilde{X}^M)$.

Importantly, $(D^1, \tilde{X}^1) \ldots, (D^M, \tilde{X}^M)$ and $(D^*, \tilde{X}^*)$ are all i.i.d. copies drawn from $P^{\text{null}}$, and thus are exchangeable. Therefore, by Theorem 3.1

$$\mathbb{E}_{(D^1, \tilde{X}^1), \ldots, (D^M, \tilde{X}^M)}\mathbb{E}_{(D^*, \tilde{X}^*)}\left[\breve{e}^j[D^j, \tilde{X}^j] \mid (D^1, \tilde{X}^1) \ldots, (D^M, \tilde{X}^M)]\right] = 1.$$

Thus,

$$\mathbb{E}[\breve{e}^j[D^j, \tilde{X}^j]] > 1, \quad 1 \leq j \leq M$$

Finally, since $\sum_{j=0}^M \mathbb{E}[\breve{e}^j[D^j, \tilde{X}^j]] = M + 1$, and $\{\mathbb{E}[\breve{e}^j[D^j, \tilde{X}^j]]\}_{j=1}^M$ are all equal,

$$\mathbb{E}[\breve{e}^0[D^0, \tilde{X}^0]] = (M+1) - M\mathbb{E}[\breve{e}^j[D^1, \tilde{X}^1]] < (M+1) - M \cdot 1 = 1,$$

as required. $\square$

Finally, it remains to show that the conditional e-statistic $\mathbb{E}_{H(\mathcal{P}_1^{\text{composite}})}[\breve{e}[D_t, \underline{\tilde{X}}_t]]$ is valid when testing a composite null, that is $\mathbb{E}[\breve{e}[D_t, \underline{\tilde{X}}_t] \mid \mathcal{F}_{t-1}] \leq 1$. However, this is immediate since given the historical data $D_{-t}$, $K_t$ is fixed, and the proof is similar to the (unconditional) finite-sample imputed e-statistic.

Next, as before, let $S_t = \prod_t \breve{e}[D_t, \underline{\tilde{X}}_t]$. It is readily seen that, assuming a label shift setting, under $H_0(\mathcal{P}_1^{\text{composite}})$, $\mathbb{E}_{H_0}[S_1] = 1$. Thus, under the null, $S_t$ is a test supermartingale:

**Corollary C.1.** *Assuming a label shift setting (concept shift), under $H_0(\mathcal{P}_1^{\text{composite}})$, $\mathbb{E}[S_t | S_1, \ldots, S_{t-1}] \leq S_{t-1}$.*

By Ville's inequality, we get type-I error control. Crucially, as before, validity holds regardless of the choice of $\mathcal{A}$, $N$, $n$, and the accuracy of predictions.

## D. Extension to Composite Null under Concept-Shift

In this section we wish to show that in the binary case, $\mathcal{P}_2$ can be extended to composite null testing:

$$(\mathcal{P}_2^{\text{composite}}) \quad H_0 : \theta_{Y|X} \leq \theta_{Y|X}^{\text{null}} \text{ for all } X = x$$
$$H_1 : \theta_{Y|X} > \theta_{Y|X}^{\text{null}} \text{ for at least one } X = x.$$

Recall that when hypothesizing about $P_{Y|X}$, for identifiability, we assume a concept-shift setting, i.e., $P_X$ is fixed. Assume $K$ is of the form $K(\underline{y}) = \sum_{i=1}^N \psi(\tilde{y}_i)$, where $\psi(\cdot)$ is a positive increasing function.

We begin by updating the definition of the population imputed e-statistic to match the composite null setting as follows:

$$e[D, \underline{\tilde{X}}] = \frac{K(\underline{\tilde{Y}})}{\mathbb{E}_{P^{\text{null}}(D, \underline{\tilde{X}})}[K(\underline{\tilde{Y}})]},$$

where $P_{XY}^{\text{null}} = P_X P_{Y|X}(; \theta_{Y|X}^{\text{null}})$. Let $(D, \underline{\tilde{X}}) \sim P = P_X P_Y(; \theta_{Y|X}) \in H_0(\mathcal{P}_2^{\text{composite}})$. We begin by showing that the population imputed e-statistic, $e[D, \underline{\tilde{X}}]$, is valid when testing a composite null:

**Under concept-shift regime, $e[D, \underline{\tilde{X}}]$ is valid when testing a composite null**

*Proof.* let $(D^*, \underline{\tilde{X}}^*) \sim P_{XY}^{\text{null}}, (D_0, \underline{\tilde{X}}^0) \sim P \in H_0(\mathcal{P}_2^{\text{composite}})$, and observe that by Theorem B.10,

$$\sum_{i=1}^N \mathcal{A}[D_0, \tilde{X}_i^0] <_{FOSD} \sum_{i=1}^N \mathcal{A}[D^*, \tilde{X}_i^*].$$

Since $K$ is strictly increasing, we then have

$$\mathbb{E}_{P(D_0, \underline{\tilde{X}}^0)}[K(\underline{\tilde{Y}})] \leq \mathbb{E}_{P^{\text{null}}(D^*, \underline{\tilde{X}}^*)}[K(\underline{\tilde{Y}})].$$

Thus, $\mathbb{E}_P\left[\frac{K(\underline{\tilde{Y}})}{\mathbb{E}_{P^{\text{null}}(D, \underline{\tilde{X}})}[K(\underline{\tilde{Y}})]}\right] \leq 1$, as needed. $\square$

Next we turn to establish the validity of the finite-sample imputed e-statistic in the composite null setting:

**Under concept-shift regime, $\breve{e}[D, \underline{\tilde{X}}]$ is valid when testing a composite null**

*Proof.* Let $(D^0, \underline{\tilde{X}}^0) = (D, \underline{\tilde{X}})$ be the observed data, and let $(D^1, \underline{\tilde{X}}^1), \ldots, (D^M, \underline{\tilde{X}}^M)$ be $M$ i.i.d datasets drawn from $P^{\text{null}}$.

Define

$$\breve{e}^j[D^j, \underline{\tilde{X}}^j] = \frac{(M+1) \sum_{k=1}^N \psi(\mathcal{A}[D^j, \tilde{X}_k^j])}{\left(\sum_{j=1}^N \psi(\mathcal{A}[D^0, \tilde{X}_j^0]) + \sum_{i=1}^M \sum_{j=1}^N \psi(\mathcal{A}[D^i, \tilde{X}_j^i])\right)},$$
$$j = 0, \ldots, M.$$

For $j \neq 0$, by the law of total expectation

$$\mathbb{E}[\breve{e}^j[D^j, \tilde{\underline{X}}^j]] = \mathbb{E}_{(D^1, \tilde{\underline{X}}^1), \dots, (D^M, \tilde{\underline{X}}^M)}[\mathbb{E}_{(D^0, \tilde{\underline{X}}^0)}\left[\breve{e}^j[D^j, \tilde{\underline{X}}^j]|(D^1, \tilde{\underline{X}}^1)\dots, (D^M, \tilde{\underline{X}}^M)\right]$$

Next, note that for fixed $(D^1, \tilde{\underline{X}}^1), \dots, (D^M, \tilde{\underline{X}}^M)$, the inner integrand of the RHS can be written as follows:

$$\mathbb{E}_{(D^0, \tilde{\underline{X}}^0)}\left[\frac{C_0}{\sum_i \psi(\mathcal{A}[D^0, \tilde{X}_i^0]) + C_1}\right] =$$

$$\mathbb{E}_{(D^0, \tilde{\underline{X}}^0)}\left[\frac{C_0}{\psi(1)\sum_i \mathcal{A}[D^0, \tilde{X}_i^0] + \psi(0)(N - \sum_i \mathcal{A}[D^0, \tilde{X}_i^0]) + C_1}\right],$$

where $C_0, C_1$ are positive constants. Since $\psi(\cdot)$ is strictly increasing, so is

$$g(s) = \psi(1)s + \psi(0)(N - s), \ \ s = 0, \dots, N.$$

Hence

$$g^*(s) = \frac{C_0}{g(s) + C_1}$$

is strictly decreasing in s. Next, let $(D^*, \tilde{\underline{X}}^*) \sim P^{\text{null}}$ independent of $(D^1, \tilde{\underline{X}}^1), \dots, (D^M, \tilde{\underline{X}}^M)$.

By Proposition B.10, $\sum_{j=1}^N \mathcal{A}[D^0, \tilde{X}_j^0] <_{FOSD} \sum_{j=1}^N \mathcal{A}[D^*, \tilde{X}_j^*]$. Thus,

$$\mathbb{E}_{(D^*, \tilde{\underline{X}}^*)}\left[\frac{C_0}{g(\sum_{j=1}^N \mathcal{A}[D^*, \tilde{X}_j^*]) + C_1}\right] < \mathbb{E}_{(D^0, \underline{X}^0)}\left[\frac{C_0}{g(\sum_{j=1}^N \mathcal{A}[D^0, X_j^0]) + C_1}\right],$$

Therefore

$$\mathbb{E}[\breve{e}^j[D^j, \tilde{\underline{X}}^j]] > \mathbb{E}_{(D^1, \tilde{\underline{X}}^1), \dots, (D^M, \tilde{\underline{X}}^M)}\mathbb{E}_{(D^*, \tilde{\underline{X}}^*)}\left[\breve{e}^j[D^j, \tilde{\underline{X}}^j] \mid (D^1, \tilde{\underline{X}}^1)\dots, (D^M, \tilde{\underline{X}}^M)]\right],$$

where the expectation on the LHS is with respect to the observed data $(D^0, \tilde{\underline{X}}^0)$ and the $M$ datasets $(D^1, \tilde{\underline{X}}^1)\dots, (D^M, \tilde{\underline{X}}^M)$.

Importantly, $(D^1, \tilde{\underline{X}}^1)\dots, (D^M, \tilde{\underline{X}}^M)$ and $(D^*, \tilde{\underline{X}}^*)$ are all i.i.d. copies drawn from $P^{\text{null}}$, and thus are exchangeable. Therefore, by Theorem 3.1

$$\mathbb{E}_{(D^1, \tilde{\underline{X}}^1), \dots, (D^M, \tilde{\underline{X}}^M)}\mathbb{E}_{(D^*, \underline{X}^*)}\left[\breve{e}^j[D^j, \tilde{\underline{X}}^j] \mid (D^1, \tilde{\underline{X}}^1)\dots, (D^M, \tilde{\underline{X}}^M)]\right] = 1.$$

Thus,

$$\mathbb{E}[\breve{e}^j[D^j, \tilde{\underline{X}}^j]] > 1, \ \ 1 \leq j \leq M$$

Finally, since $\sum_{j=0}^M \mathbb{E}[\breve{e}^j[D^j, \tilde{\underline{X}}^j]] = M + 1$, and $\{\mathbb{E}[\breve{e}^j[D^j, \tilde{\underline{X}}^j]]\}_{j=1}^M$ are all equal,

$$\mathbb{E}[\breve{e}^0[D^0, \tilde{\underline{X}}^0]] = (M + 1) - M\mathbb{E}[\breve{e}^j[D^1, \tilde{\underline{X}}^1]] < (M + 1) - M \cdot 1 = 1,$$

as required. $\qquad\qquad\qquad\qquad\qquad\qquad\qquad\qquad\qquad\qquad\qquad\qquad\qquad\qquad\square$

Finally, it remains to show that the conditional e-statistic $\mathbb{E}[\breve{e}[D_t, \tilde{\underline{X}}_t]]$ is valid when testing a composite null, that is $\mathbb{E}_{H(\mathcal{P}_2^{\text{composite}})}[\breve{e}[D_t, \tilde{\underline{X}}_t] \mid \mathcal{F}_{t-1}] \leq 1$. However, this is immediate since given the historical data $D_{-t}$, $K_t$ is fixed, and the proof is similar to the (unconditional) finite-sample imputed e-statistic.

Next, as before, let $S_t = \prod_t \breve{e}[D_t, \tilde{\underline{X}}_t]$. It is readily seen that, assuming a label shift setting, under $H_0(\mathcal{P}_2^{\text{composite}})$, $\mathbb{E}_{H_0}[S_1] = 1$. Thus, under the null, $S_t$ is a test supermartingale:

**Corollary D.1.** *Assuming a label shift setting (concept shift), under $H_0(\mathcal{P}_2^{\text{composite}})$, $\mathbb{E}[S_t|S_1, \dots, S_{t-1}] \leq S_{t-1}$.*

By Ville's inequality, we get type-I error control. Crucially, as before, validity holds regardless of the choice of $\mathcal{A}$, $N$, $n$, and the accuracy of predictions.

## E. Prediction Powered Inference: Background, Formulation, and Comparisons

### E.1. A PPI based Sequential Test

To compare our method against a test that leverages both labeled and unlabeled data, we employ PPI to design a test for the marginal of $Y$. Let $\hat{f}_t(x) \in \{0, 1\}$ be a predictor of $Y$ given $X$, learned from historical data. In our experiments, $\hat{f}$ is the bayes classifier (6) or the threshold classifier (4). Following the construction in (Einbinder et al., 2025) , given a sequence $(X_1, Y_1, \underline{\tilde{X}}_1), \dots, (X_t, Y_t, \underline{\tilde{X}}_t)$ of labeled and unlabeled data, we define the following betting procedure: At each step $t$, we bet $\lambda_t$ of our wealth against the null and receive the payoff:

$$e_{\text{PPI}}[X_t, Y_t, \underline{\tilde{X}}_t] = 1 + \lambda_t(w_t - \theta_Y^{\text{null}}), \tag{9}$$

where $\lambda_t \in [-1/2, 1/2]$ and $w_t$ is the PPI estimator for the mean of $Y$:

$$w_t = Y_t + \epsilon_t \left( \frac{1}{N} \sum_{X \in \underline{\tilde{X}}_t} \hat{f}_t(X) - \hat{f}_t(X_t) \right).$$

Thus, the resulting e-process is $E_T^{\text{PPI}} = \prod_{t=1}^T e_{\text{PPI}}[X_t, Y_t, \underline{\tilde{X}}_t]$.

To maximize the power of the PPI process we adaptively set both $\epsilon_t$ and $\lambda_t$ as follows: $\epsilon_t$ is used to reduce the variance of the PPI estimator $w_t$. Indeed, following (Angelopoulos et al., 2023b), we minimize the variance of the PPI estimator by setting $\epsilon_t$ as follows:

$$\epsilon_t = \frac{\widehat{\text{COV}}(\underline{Y}_{-t}, f_t(\underline{X}_{-t}))}{\left(1 + \frac{1}{N}\right) \widehat{\text{VAR}}(f_t(\tilde{X}_{-t}))}, \tag{10}$$

where $(\underline{X}_{-t}, \underline{Y}_{-t}), \tilde{X}_{-t}$ are the sequences of labeled and unlabeled data observed up to time $t$, and $\widehat{\text{COV}}$ and $\widehat{\text{VAR}}$ are the estimates of the covariance and variance respectively. To set the bets $\{\lambda_t : t \geq 1\}$ we use online newton step (ONS):

$$\lambda_t = \min \left\{ \frac{1}{2}, \max \left\{ -\frac{1}{2}, \lambda_{t-1} + \frac{2}{2 - \log 3} \frac{z_t}{a_t} \right\} \right\},$$

where

$$z_t = \frac{w_t}{(1 + w_t \lambda_{t-1})}$$
$$a_{i,t} = a_{t-1} + z_t^2$$

By using ONS, we ensure the exponential growth of the e-process $E_T^{\text{PPI}}$ (Shekhar & Ramdas, 2023).

The validity of the PPI estimator follows immediately from the fact that $w_t$ is an unbiased estimator of $\mathbb{E}_{H_0(\mathcal{P}_1)}[Y_t]$:

**Proposition E.1.** $e_{PPI}[X_t, Y_t, \underline{\tilde{X}}_t]$ *is a valid sequential e-value against* $H_0(\mathcal{P}_1)$.

*Proof.* For every $t$ and for every $\tilde{X} \in \underline{\tilde{X}}_t$ we have that $X_t \overset{d}{=} \tilde{X}$. Furthermore, since $\hat{f}_t(X)$ is a function of the historical data only, we have that:

$$\mathbb{E}_{H_0(\mathcal{P}_1)} \left[ \frac{1}{N} \sum_{\tilde{X} \in \underline{\tilde{X}}_t} \hat{f}_t(\tilde{X}) - \hat{f}_t(X_t) \mid \mathcal{F}_{t-1} \right] = 0. \tag{11}$$

Therefore,

$$\mathbb{E}_{H_0(\mathcal{P}_1)}[e_{\mathrm{PPI}}[X_t, Y_t, \tilde{\underline{X}}_t] \mid \mathcal{F}_{t-1}] = \mathbb{E}_{H_0(\mathcal{P}_1)}[1 + \lambda_t(w_t - \theta_0) \mid \mathcal{F}_{t-1}] =$$

$$\mathbb{E}_{H_0(\mathcal{P}_1)}\left[1 + \lambda_t\left(Y_t + \epsilon_t\left(\sum_{\tilde{X} \in \tilde{\underline{X}}_t} \hat{f}_t(\tilde{X}) - \hat{f}_t(X_t)\right) - \theta_0\right) \mid \mathcal{F}_{t-1}\right] \overset{(1)}{=}$$

$$1 + \lambda_t\mathbb{E}_{H_0(\mathcal{P}_1)}[Y_t \mid \mathcal{F}_{t-1}] + \lambda_t\epsilon_t\mathbb{E}_{H_0(\mathcal{P}_1)}\left[\sum_{\tilde{X} \in \tilde{\underline{X}}_t} \hat{f}_t(\tilde{X}) - \hat{f}_t(X_t) \mid \mathcal{F}_{t-1}\right] - \lambda_t\theta_0 \overset{(2)}{=}$$

$$1 + \lambda_t\mathbb{E}_{H_0(\mathcal{P}_1)}[Y_t \mid \mathcal{F}_{t-1}] - \lambda_t\theta_0 \overset{(3)}{=} 1,$$

Where, (1) holds since $\lambda_t$ and $\epsilon_t$ are functions of the historical data only, (2) stems from (11), and (3) is true since the labeled samples $Y_t$ are i.i.d and under the null $\lambda_t\mathbb{E}_{H_0(\mathcal{P}_1)}[Y_t] = \theta_0$.

Thus, $\mathbb{E}_{H_0(\mathcal{P}_1)}[e_{\mathrm{PPI}}[X_t, Y_t, \tilde{\underline{X}}_t] \mid \mathcal{F}_{t-1}] = 1$ and the proof is complete.

$\square$

### E.2. Analyzing the Benefit of PPI

To understand the benefit of using PPI compared to solely using labeled data, we compare PPI to a sequential test that uses the same e-statistic as PPI but with $\epsilon = 0$.

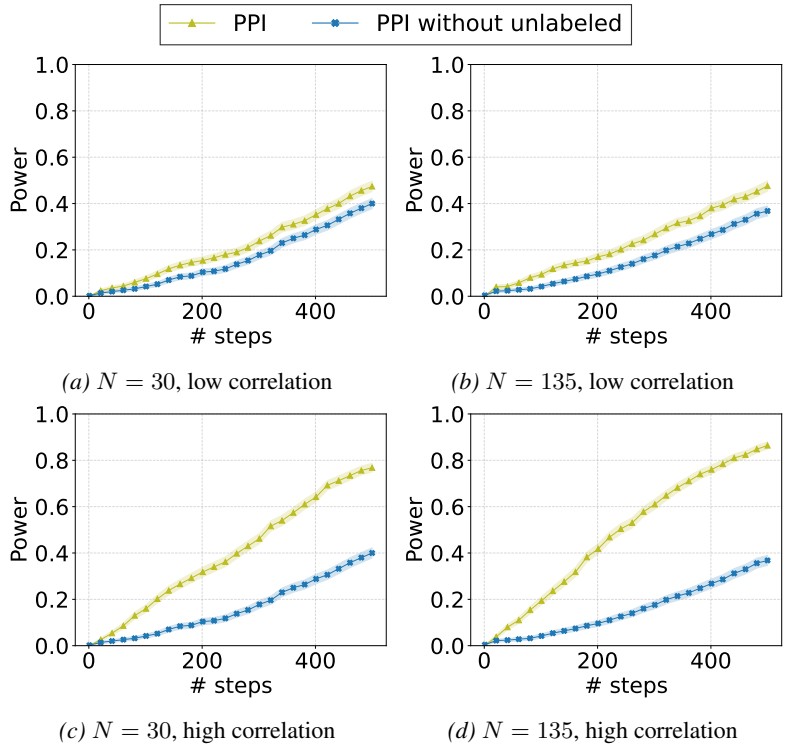

*(a) $N = 30$, low correlation*     *(b) $N = 135$, low correlation*

*(c) $N = 30$, high correlation*     *(d) $N = 135$, high correlation*

*Figure 5.* Comparing PPI to only using labeled data for the label shift setting of Section 5.4.

### E.3. One-Sided PPI

The PPI e-process, as we defined in (9), is a two-sided test. However, our setting is one-sided, and while it is valid to use a two-sided test for a one-sided setup, it places the two-sided test at a disadvantage in terms of power. To simulate a fair comparison, we modify the PPI e-process to be aware of the one-sided setup by forcing $\lambda_t$ to be non-negative, and we refer

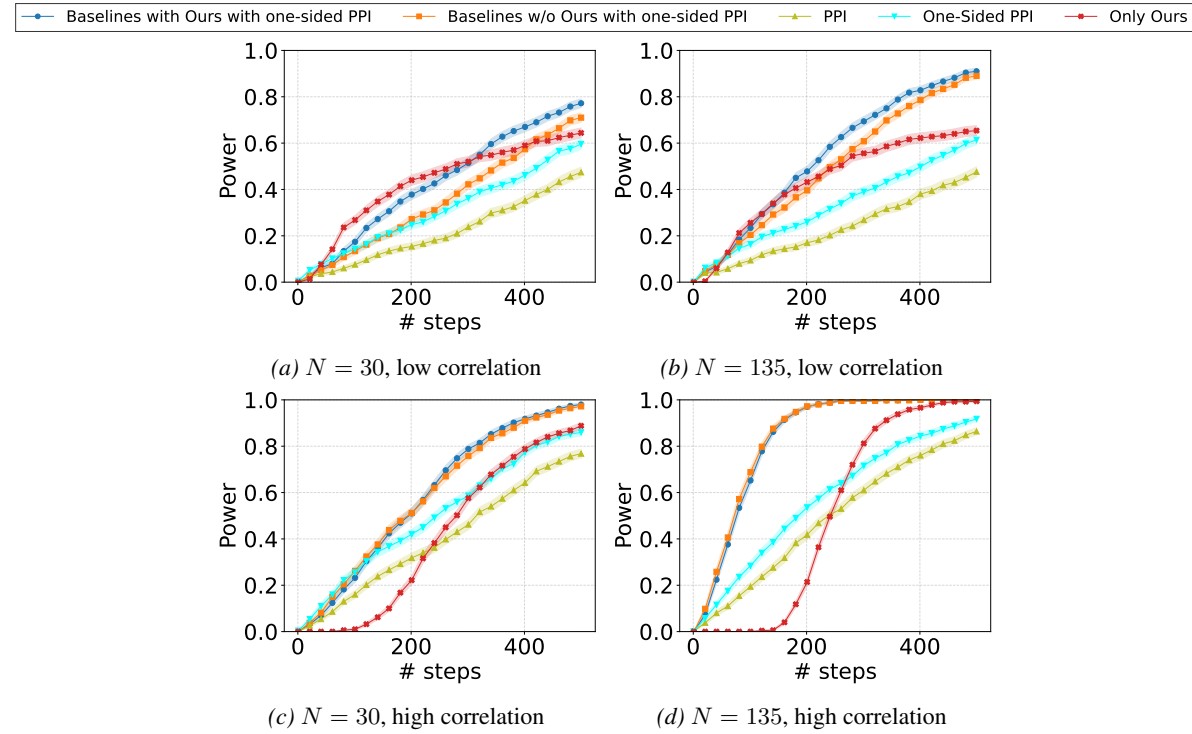

*Figure 6.* Comparing the imputed process to PPI and one-sided PPI. We simulate a small and large number of unlabeled points per batch ($N = 35$ and $N = 135$, respectively), and high correlation (0.3) and low correlation (0.7) settings.

to this process as the one-sided PPI e-process. To validate our method against the new process, we repeat the simulation setup of Section 5.4, but now we use the one-sided PPI e-process in the baselines. Figure 6 depicts the results. We can see that the one-sided PPI is indeed more powerful than the PPI test. When comparing the convex combination of the baselines that includes the one-sided PPI with the convex combination that also includes the imputed e-process, we can see that the imputed e-process is still able to collect evidence against the null that is not used by the one-sided PPI and ultimately improves the power of the test.

### E.4. Comparison to Our approach

The major difference between our method and PPI is how the correlation between $X$ and $Y$ affects each test. When the correlation is low, the resulting predictive model $\hat{f}_t$ may be poor in the sense that the covariance of the prediction, $f_t(X)$, and the labels, $Y$, may be low. In this situation, as can be seen from (10), the PPI test pushes $\epsilon$ to zero, which boils down to a test that utilizes only labeled data. Conversely, the power of the imputed process stems from applying $\mathcal{A}$ on both the null and observed data and comparing the distribution of pseudo labels. Thus, even when the correlation is low, and thus the classifier is of low accuracy, the imputed e-process can still detect the divergence between the null and alternative. Indeed, we can see in Figures 5a and 5b that when the correlation is low, the PPI process is only marginally better than a test that only uses labeled data and under the same settings we can see in Figures 6a and 6b that the imputed e-process is significantly better than both the PPI and one sided PPI e-process.

## F. Simulations

### F.1. Label Shift: Simulation Setup

For the null hypothesis, we set $\theta_Y^{\text{null}} = 0.5$ and for the alternative, we use $\theta_Y = 0.52$. For the low correlation setting we set $P(X = 1|Y = 0) = 0.35$ and $P(X = 1|Y = 1) = 0.65$. For the high correlation setting we use $P(X = 1|Y = 0) = 0.2$ and $P(X = 1|Y = 1) = 0.9$.

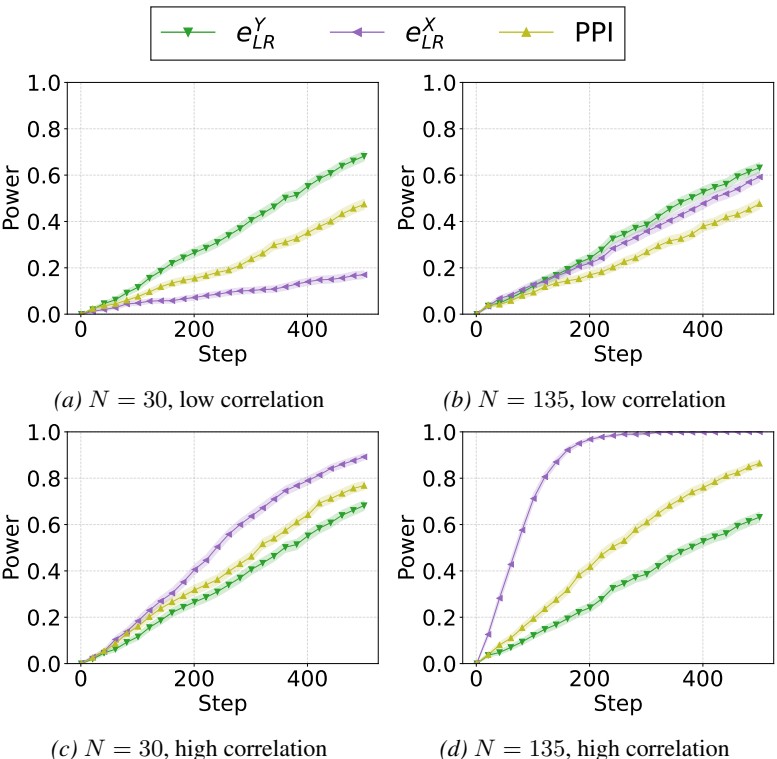

*Figure 7.* Power curve of each baseline used in the simulations from section 5.4.

## F.2. Concept Shift: Simulation Setup

We set $\theta_{Y|X=0}^{\text{null}} = 0.4$, $\theta_{Y|X=1}^{\text{null}} = 0.7$, and $\theta_X^{\text{null}} = 0.5$ for the low correlation setting and $\theta_{Y|X=0}^{\text{null}} = 0.2$, $\theta_{Y|X=1}^{\text{null}} = 0.85$, and $\theta_X^{\text{null}} = 0.5$ for the high correlation setting. To sample from a distribution from the alternative we use: $\theta_{Y|X=0} = \theta_{Y|X=0}^{\text{null}} + 0.02$ and $\theta_{Y|X=1} = \theta_{Y|X=1}^{\text{null}} + 0.02$.

## F.3. Hyperparameter Tuning

We repeat the low $N$, low correlation simulations of Section 5.4 with different values of $M$. Figure 9 shows the results. It is evident that increasing $M$ yields a monotonic improvement in the power of the method. In addition, we can see that for $M > 32$, the improvement is only asymptomatic.

## F.4. Concept Shift: Non Monotone Hypothesis

To demonstrate that the power of the test against the alternative is indeed tied to the choice of $K$, we repeat the same simulation as in Section 5.5, but we sample from an alternative that is not right-sided. Specifically, we set $\theta_{Y|X=0}^{\text{null}} = 0.4$, $\theta_{Y|X=1}^{\text{null}} = 0.7$, and $\theta_X^{\text{null}} = 0.5$ for the low correlation setting and $\theta_{Y|X=0}^{\text{null}} = 0.2$, $\theta_{Y|X=1}^{\text{null}} = 0.85$, and $\theta_X^{\text{null}} = 0.5$ for the high correlation setting. To sample from a distribution from the alternative we use: $\theta_{Y|X=0} = \theta_{Y|X=0}^{\text{null}} + 0.02$ and $\theta_{Y|X=1} = \theta_{Y|X=1}^{\text{null}} - 0.02$. Figure 10 depicts the results. We can observe that indeed our method does not perform well under this setting. Notably, all of the baselines perform poorly as well.

# G. LLM Evaluation

## G.1. Experimental Setup

In this section, we provide a full description of the datasets, the parameters, and the setup of the experiments.

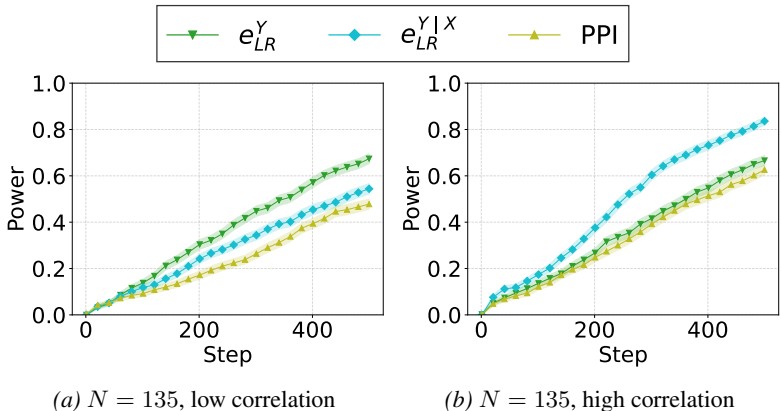

*(a)* $N = 135$, low correlation    *(b)* $N = 135$, high correlation

*Figure 8.* Power curve of each baseline used in the simulations from section 5.5.

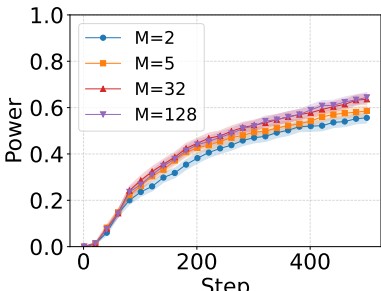

*Figure 9.* The effect of $M$ on the power of the imputed e-process in the label shift setting from section 5.4.

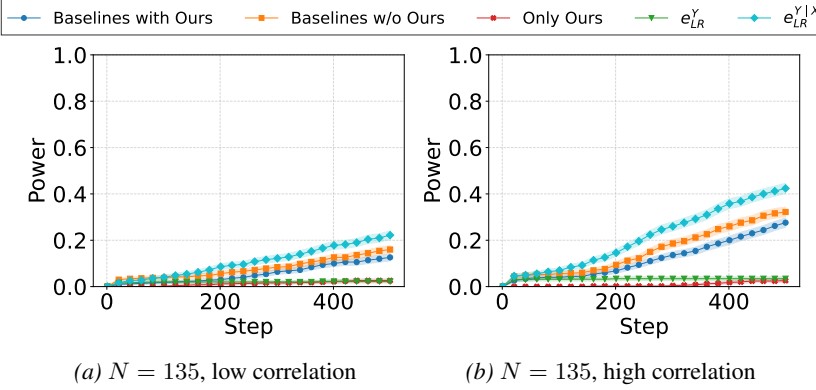

*(a)* $N = 135$, low correlation    *(b)* $N = 135$, high correlation

*Figure 10.* Power curves for testing an alternative hypothesis which is not right sided, i.e., $\theta_{Y|X=0} > \theta_{Y|X=0}^{\text{null}}$ and $\theta_{Y|X=1} < \theta_{Y|X=1}^{\text{null}}$.

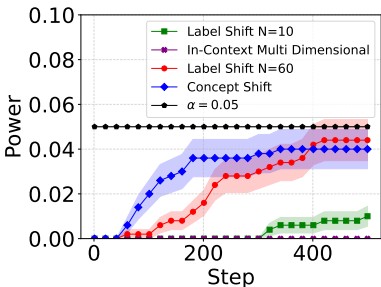

*Figure 11.* False rejection rate for real-world experiments for $\alpha = 0.05$

### G.2. Data

GSM8K, which contains 87,900 grade school math problems. MATH (Hendrycks et al., 2021), which is a dataset of 12,500 challenging math problems, and AQUA-RAT (Algebra Question Answering with Rationals) (Ling et al., 2017), which contains about 100,000 algebra questions.

### G.3. Concept Shift

To sample from the null distribution we use: $\theta_{Y|X=0}^{\text{null}} = 0.8$, $\theta_{Y|X=1}^{\text{null}} = 0.5$ and $\theta_Y^{\text{null}} = 0.55$. To sample from a distribution from the alternative, we use: $\theta_{Y|X=0} = 0.8$, $\theta_{Y|X=1} = 0.535$, and $\theta_Y = 0.58$. After processing the data, we are left with 42425 samples for model $A$ and 50909 samples for model $B$.

### G.4. Label Shift

To sample from the null distribution we use: $\theta_Y^{\text{null}} = 0.58$ and $\theta_X^{\text{null}} = 0.55$. To sample from a distribution from the alternative, we use: $\theta_Y = 0.594$ and $\theta_X = 0.557$. After processing the data, we are left with 176162 samples for model $A$ and 238026 samples for model $B$.

### G.5. Validity

We repeat the real-world experiments in section 5, but now model $A$ and model $B$ are the same. In this setting, the null is true, and we expect that the rejection rate will not exceed $\alpha$. Similarly, we repeat the experiment in section 7 but we use the data from the same year for both the null and observed data. Figure 11 shows the results. We can see that the false rejection rate of all tests was less than $\alpha$.

## H. In-Context Learning

Census data was obtained through the folktables interface (Ding et al., 2021).

The full list of features used is:

- Sex

- Disability

- Employment status of parents

- Military service

- Nativity

- Hearing difficulty

- Vision difficulty

- Cognitive difficulty

- Recoded detailed race code

- Total person's income

