# OpenReview forum: "Semi-Supervised Hypothesis Testing by Betting on Predictions"
_ICML.cc/2026/Conference — ICML 2026 regular_

### Official Review · Reviewer_1HGi · 2026-03-07

**Soundness:** 4
**Presentation:** 2
**Significance:** 4
**Originality:** 3
**Overall Recommendation:** 5
**Confidence:** 3

**Summary:**

The submission addresses sequential hypothesis testing in semi-supervised settings, aiming to accelerate test stopping times while strictly controlling anytime-valid Type 1 error. The authors construct a finite-sample “imputed e-statistic” that leverages predictions from an arbitrary machine learning model trained on the small labeled batch and applied to the large unlabeled batch.

The authors formally establish the identifiability conditions (label shift and concept shift) required for valid marginal and conditional inference, while also proving a Total Variation robustness bound where the null distribution is only empirically approximated. For binary data, they prove this e-statistic yields non-trivial power $\mathbb{E}_{H_1} [e] > 1$, ensuring the resulting e-process acts as a submartingale that will eventually reject the null.

Simulations and a real-world application to LLM evaluation are shown, where it is demonstrated that the proposed method achieves significantly faster stopping times compared to sequential likelihood ratio tests and prediction-powered inference, particularly when the correlation between covariates and labels is weak.

**Compliance With Llm Reviewing Policy:**

Affirmed.

**Key Questions For Authors:**

1. The experiments use $M = 128$, which requires training/predicting $\mathcal{A}$ 129 times per sequential batch, and the authors note that this is feasible for simple models. Can online updating or model-reuse schemes be used for $\mathcal{A}$ that preserve the exchangeability guarantees while mitigating this $\mathcal{O}(M)$ training cost?

2. Is it possible to provide an upper bound on the expected stopping time that explicitly characterizes how the power scales with the number of unlabeled samples $N$, the quality of the predictive model $\mathcal{A}$, the divergence/distance $D(P_1 \| P_0)$, or $M$? The lack of quantitative rates makes it difficult to gauge the efficiency gains over standard fixed-sample $p$-values.

**Limitations:**

Yes

**Strengths And Weaknesses:**

**Soundness**

* [Strength] The proposed framework guarantees valid anytime type-1 error control. If the learned model is weak or the correlation is low, the e-statistic concentrates around 1 (stalling the test), rather than inflating the false-positive rate. This robustness makes the statistical soundness of the test entirely decoupled from the empirical risk of the neural network.

* [Weakness] The formal performance guarantees (Theorems 4.1 and 4.2) are purely qualitative. The result proves non-trivial power, which merely guarantees that the expected stopping time is finite. There is no characterization of how the expected stopping time scales with the margin of the classifier, the dimension of $X$, or the “distance” between $P_0$ and $P_1$.

**Presentation**

* [Strength] The paper is generally well written, clearly formalizing the required identifiability conditions and explaining why testing exchangeability on $X$ allows one to draw conclusions about $Y$.

* [Weakness] The implementation of computing the estimator in Equation (2) can be better explained. The paper does not make it immediately obvious that sampling $M$ null datasets requires retraining the predicting algorithm $\mathcal{A}$ from scratch $M + 1$ times at every single sequential step. It is also not obvious that the $M$ null datasets all require the same $n$ labeled and $N$ unlabeled points.

* [Weakness] The title is under-descriptive. My understanding is that this paper’s key contribution is in using unlabeled data to accelerate sequential testing. The paper’s visibility could be improved by modifying the title so that it's clear that the paper is about semi-supervised inference or A/B testing.

**Significance**

* [Strength] The application to LLM evaluation is highly relevant. Generating text $X$ is virtually free while human grading $Y$ is expensive, and this paper provides a method to stop A/B tests early by leveraging cheap unlabeled data.

* [Weakness] The proposed method is computationally expensive. The predictive model $\mathcal{A}$ must be trained $M + 1$ times per step from scratch, and so the framework is restricted to computationally trivial models (e.g., linear classifiers; zero-shot API calls).


**Originality**

* [Strength] Merging the finite-sample exchangeability trick from conformal prediction with the e-process literature is an original methodological contribution.

---

> ### Author Rebuttal · Authors · 2026-03-28
>
> We sincerely thank the reviewer for the highly positive assessment of our work. We are thrilled that you found the mathematical soundness and the relevance to LLM evaluation to be significant strengths. We address your insightful critiques and questions below.
>
> > Presentation of implementation details of the imputed e-statistics
>
> we thank the reviewer for the valuable comment. We will clarify these important details in the main text body.
>
> > The title
>
> we agree with the reviewer that the paper’s visibility could be improved by explicitly calling out the fact that our testing approach is semi-supervised by nature. The new title is now: “Semi-Supervised Hypothesis Testing by Betting on Predictions”.
>
> > Is it feasible to use complex models? Can online updating and model re-use schemes be used to mitigate the $O(M)$ training cost?
>
> In simulations we show that the method can be  powerful even when $M$ is small ($M=5$), as can be seen in Figure 8 in Appendix F3. Additionally, it is feasible to use our method with complex models via in-context learning, as we demonstrated in a new experiment, see our response to reviewer gfYf under “New Experiment: In-Context Learning with High Dimensional Mixed Type Data”, in this case we used $M=2$, further demonstrating that favorable results can be obtained with a low value of $M$. A related question was also raised by reviewer 3MXb, in the interest of space we kindly refer the reviewer to our answer “End-to-End Computational Complexity & Baselines” for further details.
>
> > Upper bounding the stopping time
>
> Thank you for raising this important issue. We agree that this is indeed an important research direction. Beyond the fact that it will characterize the influence of $\mathcal{A}$, $M$ and $K$ on the power of the test, such analysis will providing guiding principles for the design of the test. Naturally, answering this question requires further research that we intend to explore in future work.
>
> We thank the reviewer for their valuable questions and we will add the appropriate discussions on the applicability of our method to complex models and future theoretical work.

---

> > ### Author Rebuttal · Reviewer_1HGi · 2026-04-01
> >
> > Thank you for the response. I am satisfied with the answers and will maintain my score.

---

### Official Review · Reviewer_Lece · 2026-03-08

**Soundness:** 1
**Presentation:** 1
**Significance:** 1
**Originality:** 1
**Overall Recommendation:** 1
**Confidence:** 3

**Summary:**

The paper aims at improving the performance of supervised machine learning via unlabelled data. The authors calls the approach “betting on predictions”. Conceptually the approach reminds of semi-supervised learning in the classical machine learning.
The paper introduces a lot of formalisms, which seem to be somewhat disconnected from the empirical section. Interpetations of the experimental results are relatively shallow and the implications of the proposed approach to machine learning in general are not entirely clear.

**Compliance With Llm Reviewing Policy:**

Affirmed.

**Key Questions For Authors:**

What are the main interpetations of the formal analysis and the experimental results?

**Limitations:**

The formal and the experimental analysis are relatively disconnected. Implications to machine learning in general are not entirely clear.

**Strengths And Weaknesses:**

Strengths:
-	Extensive experimental work
-	Formal treatment
Weaknesses
-	Formalisms are somewhat disconnected from the empirical analysis
-	Conclusive interpetations of the theoretical and experimental analysis are lacking
-	Implications of the analysis to machine learning in general are not clear

---

> ### Author Rebuttal · Authors · 2026-03-28
>
> We thank the reviewer for their time. However, we respectfully note a fundamental misunderstanding regarding the core focus of our paper. Our work does not aim to improve supervised machine learning. Instead, we introduce a framework that leverages predictions on unlabeled data to enhance the power of sequential hypothesis tests.
> The theoretical results of our work mathematically prove that our proposed test is statistically valid and achieves non-trivial power. The empirical section directly validates these guarantees, demonstrating that our method outperforms strong baseline methods. Beyond the fact that we use machine learning methods to improve the power of hypothesis tests by leveraging unlabeled data, we also demonstrate the applicability of our method to LLM evaluations using judge annotations among other applications.
> In light of this clarification, we respectfully ask the reviewer to reasses our work. We will be happy to engage in conversation during the discussion period and provide additional clarifications and details.

---

> > ### Author Rebuttal · Reviewer_Lece · 2026-04-03
> >
> > Thanks for the comment. I keep my opinion regarding limited contribution of the findings. This is not easily fixable by revisions. For the area chair: please disregard my opinion if the majority thinks otherwise.

---

### Official Review · Reviewer_3MXb · 2026-03-09

**Soundness:** 3
**Presentation:** 3
**Significance:** 3
**Originality:** 3
**Overall Recommendation:** 4
**Confidence:** 2

**Summary:**

The paper introduces a testing-by-betting framework that utilizes predictions on unlabeled data to increase the power of sequential hypothesis testing. By proposing an imputed e-statistic, a sequential test can be devised with provable non-trivial power even when the underlying predictive models are inaccurate.

**Compliance With Llm Reviewing Policy:**

Affirmed.

**Final Justification:**

The authors have addressed my concerns. However, I am not an expert on this domain.

**Key Questions For Authors:**

1. Can this framework be extended to non-binary data (multi-class classification or even continuous)?
2. What is the end-to-end computational complexity? How does it compare to other baselines?
3. Why do you have to assume the accuracy of the LLM judge is the same for outputs generated by two candidates in Sec 6.3? This assumption feels quite strong but I'm not exactly following here. If it is indeed necessary, how should practitioners assess whether the needed assumptions are met in real data?

**Limitations:**

yes

**Strengths And Weaknesses:**

Strengths:
1. The authors studied a fairly influential problem in a low-data setting and derived mathematically rigorous results that are also practically relevant.
2. The proposed methods can be easily combined (with a specific procedure) with existing baselines to yield better empirical test power.

Weaknesses:
1. The method requires fitting a prediction model on each fresh batch of data, which is possibly computationally difficult.

---

> ### Author Rebuttal · Authors · 2026-03-28
>
> We sincerely thank the reviewer for the positive assessment and for highlighting the mathematical rigor and practical relevance of our work
>
> > Extension to non-binary data
>
> Yes it is possible to use our method in this more general setting. This is because the validity (anytime type-I error control) of the proposed test holds even when $Y$ or $X$ are non-binary (Propositions 3.1 and 3.2). For example, our method can be used to test if a continuous reward, $Y$, has increased between model A and model B. We believe that our method should be powerful in this setting as well as long as the underlying distribution of $Y$ is stochastically ordered. This intuition is rooted in our non-trivial power results in the case of a binary $Y$. In this case we utilized the fact that the Bernoulli family is stochastically ordered, and furthermore, this stochastic order is preserved under conditioning $Y|X=x$. These properties hold for many popular distribution families, such as Normal, Exponential, and Poisson, to name a few. Having said that, formally establishing such general power analysis requires a significant extension to our theory. This is because the analysis depends on several design choices, such as the learning algorithm $A$ and the function $K$. We view such generalization as a promising next step which we currently explore. Finally, we note that the method can be applied to a non-binary, multivariate $X$, and we added a new experiment that demonstrates this (see details in our response to reviewer gfYf under “New Experiment: In-Context Learning with High Dimensional Mixed Type Data”).
>
> > End-to-End Computational Complexity & Baselines
>
> The overall complexity of our method is dominated by $M+1$ training calls on n labeled data points, and additional $M+1$ prediction steps on $N$ unlabeled data points. However, practically training complexity does not limit the applicability of our method for two key reasons:
> Even for a small $M=5$, we demonstrate in Figure 8 in Appendix F3, that the power is comparable to larger values such as $M=128$. Furthermore, in the new multivariate $X$ experiment our method outperforms all baselines for a minimal choice of $M=2$.
> Since in our setup data is scarce, fitting a complex model from scratch is not a common practice. In such setting the training algorithm is often a lightweight procedure. Yet, lightweight procedures can be powerful. As we show in the paper, even in complex problems such as LLM comparison, one can use a simple Naive Bayes model with a single feature $X$ - a judge annotation. Additionally,  in-context learning is a lightweight procedure which enables us to leverage expressive foundation models as demonstrated in the new multivariate-$X$ experiments detailed in the response to reviewer gfYf.
> In terms of comparison to PPI: In its standard form, at each step, it trains a model on the whole history. Therefore, its complexity depends on the specific choice of learning algorithm $A$ and if it can be trained in an online manner. While the complexity of PPI is lower than ours, we believe that the significant power gains demonstrated in our experiments justify the added complexity.
>
> > The LLM Judge Assumption (Section 6.3)
>
> For label-shift to hold we need that $P(X|Y)$ will be fixed between the null (model A) and alternative (model B) (i.e. $P(X_A | Y_A) = P(X_B | Y_B)$). Consider $P(X_A=1|Y_A=1)$, which is the probability that the judge has determined that model A’s  answer is correct ($X_A=1$)  given that the answer is indeed correct ($Y_A=1$). Thus, $P(X_A|Y_A)$ reflects the accuracy of the judge relative to model A, and similarly $P(X_B|Y_B)$ reflects the accuracy of the judge relative to model B. Therefore, if the accuracy of the judge is fixed then $P(X_A | Y_A) = P(X_B | Y_B)$. In this specific application, this assumption is essential to guarantee that any observed shift in estimated accuracy can be attributed to a shift in the model accuracy, and not due to a bias in the judge towards a specific model. Importantly, this assumption can be tested during the execution of the test using historical data. Specifically, the accuracy of the judge can be evaluated and compared between model A and model B. For instance, in this specific example, one can use a standard proportion test.
>
> We thank the reviewer for their thoughtful questions and we will include in the paper discussions about extension to non-binary data, the overall complexity, and the LLM judge assumption.
>
> We hope that our detailed responses and the new in-context learning experiment fully address your concerns, and we will be happy to address any remaining concerns.

---

> > ### Author Rebuttal · Reviewer_3MXb · 2026-03-31
> >
> > Thank you for addressing my questions. I have read your response and decide to maintain my score.

---

### Official Review · Reviewer_gfYf · 2026-03-13

**Soundness:** 4
**Presentation:** 3
**Significance:** 3
**Originality:** 3
**Overall Recommendation:** 5
**Confidence:** 4

**Summary:**

This paper outlines fairly rigorously a statistical approach to sequential hypothesis testing in the semi-supervised settings, with clearly demonstrated applications to evaluating LLMs when generations are cheap but human annotations are costly.

**Compliance With Llm Reviewing Policy:**

Affirmed.

**Final Justification:**

This is a very solid paper with useful theoretical analysis of sequential testing, particularly relevant for semi supervised settings (and by extension settings where high quality human labels are costly, but lower quality LLM generated labels are cheap, for e.g. llm-as-a-judge evaluations).

The authors do a good job of showing how their theory can be applied to settings like evaluation, under two important regimes - label shift and concept shift - and offer useful guidance on how to address these two settings for e.g. by using few-shot examples, pre-trained models etc.

**Key Questions For Authors:**

The LLM evaluation + LLM as a judge framing is quite useful - how do things change when I don't individually fit each batch of labelled data, but use a general purpose prediction model with good performance (e.g. a reward model/LLM judge fine-tuned on some initial batch of labels) - does absolute performance not matter as long as accuracy(A) = accuracy(B)?

**Limitations:**

yes

**Strengths And Weaknesses:**

The theory is very well-written and empirical demonstrations seem to demonstrate increased power of the author's approach. The paper's framing of the semi-supervised setting and its applicability to LLM evaluation is very clear. The empirical simulations help readers immediately connect the approach to LLM evals.

The clarity/immediacy of the method's impact could be improved by making the example more front-and-centre and spending some more time on it, as some of the theory may be hard to approach for more applied readers. Note: the /mathcal{A} notation for the predictive model is confusing given that model "A" is one of the models under evaluation.

---

> ### Author Rebuttal · Authors · 2026-03-28
>
> We sincerely thank the reviewer for their positive feedback regarding the clarity of our framing and the strength of our theoretical and empirical results.
>
> > Regarding using a fixed model without fitting it to each batch
>
>
> * In the label-shift regime, the answer is yes: it is possible to use a fixed pre-trained model $f(X)$. To see why, note that under this setting, $X$ and $Y$ are correlated and thus, their marginal distributions move together. Therefore, even if the model is fixed, any change in the distribution of $Y$ is reflected in a corresponding shift in $X$, and thus also in $f(X)$.
> * In the concept shift setting, however, the situation is different. The distribution of $X$ is the same between the null and alternative. Therefore, for a fixed $f$, the distribution of $f(X)$ is identical between the null and alternative, and does not provide any new evidence against the null. This emphasizes our contribution that captures a shift in $Y|X$ by modeling the conditional distribution under the null and alternative.
> * With that said, even under concept shift we can sill utilize the capabilities of a pre-trained model. The idea is to use in-context learning, providing the model with the labeled datapoints in its context window. This technique is appealing since it can effectively model $Y|X$ with relatively few data points. We added a new experiment that demonstrates this in-context learning approach (see details below). As can be seen, this technique allows us to enjoy the benefit of a powerful foundation model for tabular data (TabFPN[1]) to obtain a significant increase in power.
>
> > Improving the clarity of the paper
>
> We appreciate the constructive suggestions for improving readability.  In our revised manuscript, we will further elaborate on how our method can be used for LLM evaluation. We have also updated our notation, replacing “model A” and “model B”  with $m_A$ and $m_B$.
>
> ---
>
> ## New Experiment: In-Context Learning with High Dimensional Mixed Type Data
>
> In this experiment, we demonstrated that our framework:
> * Can leverage expressive pre-trained models via in-context learning.
> * Generalizes to diverse data types and multi-dimensional feature spaces.
>
> ### Experiment Setup
>
> We evaluate the procurement of private health insurance using California census data (2017–2019). We define the covariate $X$ as a multi-dimensional vector containing a mix of continuous and multiclass variables. The target variable $Y$ remains a binary indicator of private healthcare coverage.
>
> We integrate TabPFN [1], a state-of-the-art tabular foundation model. At each step, we provide TabPFN with the labeled data directly in its context window to perform inference on the new batch of unlabeled data. By pairing this in-context inference with a small number of null datasets, $M=2$, we leverage the high expressivity of a pre-trained model while maintaining a strictly low computational overhead.
>
> ### Results
> As shown in the following [link](https://anonymous.4open.science/r/betting_on_predictions_rebuttal/multivariate_health_care.pdf), we can see that:
> * Our method is the most powerful method, compared to PPI and likelihood ratio test (LRT).
> * When combining our method with PPI and LRT, we gain a significant increase in power compared to the combination of PPI and LRT.
>
> ### Additional Details
>
> 1. We ran the experiment using 10 different realizations of the data and plotted the mean power and standard error
> 2. Number of labeled samples per-batch n=50, unlabeled samples per batch N=100,  null datasets M=2.
> 3. Full List of Features Used:
> * Sex
> * Disability
> * Employment status of parents
> * Military service
> * Nativity
> * Hearing difficulty
> * Vision difficulty
> * Cognitive difficulty
> * Recoded detailed race code
> * Total person's income
>
> We have updated the manuscript and appendix to include this experiment and clarify these capabilities.
>
>
> [1] - N. Hollmann, S. Müller, K. Eggensperger, and F. Hutter. Tabpfn: A transformer that solves
> small tabular classification problems in a second. arXiv preprint arXiv:2207.01848, 2022

---

> > ### Author Rebuttal · Reviewer_gfYf · 2026-04-03
> >
> > Thanks to the authors for the response and additional analysis! it's worthwhile reiterating the difference between the label shift/concept shift regime in the LLM eval section, as it's an important one for practitioners to keep in mind (and the few shot solution seems reasonable, and can help motivate current standard practices).

---

### Decision · Program_Chairs · 2026-04-30

**Decision:**

Accept (regular)

**Comment:**

While Reviewer Lece recommended a Strong Reject, their assessment appears to be rooted in a fundamental misunderstanding of the paper's primary objective. The reviewer evaluated the submission through the lens of classical semi-supervised learning aimed at improving predictive model performance, whereas the paper is explicitly focused on improving the statistical power of sequential hypothesis tests.

The paper offers a strong, novel contribution to probabilistic methods and statistical inference in AI. I encourage the authors to include the promised discussions on stopping time bounds and the new TabPFN experiments in the final camera-ready version.